# Catalytic Neutralization of Water Pollutants Mediated by Dendritic Polymers

**DOI:** 10.3390/nano12030445

**Published:** 2022-01-28

**Authors:** Michael Arkas, Ioannis Anastopoulos, Dimitrios A. Giannakoudakis, Ioannis Pashalidis, Theodora Katsika, Eleni Nikoli, Rafael Panagiotopoulos, Anna Fotopoulou, Michail Vardavoulias, Marilina Douloudi

**Affiliations:** 1Demokritos National Centre for Scientific Research, Institute of Nanoscience and Nanotechnology, 15341 Athens, Greece; theodorakatsika@gmail.com (T.K.); h.nikoli@inn.demokritos.gr (E.N.); rafail.panagiotopoulos@hotmail.gr (R.P.); a.fotopoulou99@hotmail.com (A.F.); 2Department of Agriculture, University of Ioannina, UoI Kostakii Campus, 47040 Arta, Greece; anastopoulos_ioannis@windowslive.com; 3Department of Chemistry, Aristotle University of Thessaloniki, 54124 Thessaloniki, Greece; dagchem@gmail.com; 4Environmental & Radioanalytical Chemistry Lab, Department of Chemistry, University of Cyprus, Nicosia 1678, Cyprus; pspasch@ucy.ac.cy; 5PYROGENESIS S.A., Technological Park 1, Athinon Avenue, 19500 Attica, Greece; mvardavoulias@pyrogenesis-sa.gr

**Keywords:** dendritic polymers, dendrimers, metal nanoparticles, photocatalysis, water purification, dye discoloration, pollutant degradation, nanoparticle catalysis, decomposition, semiconductors

## Abstract

Radially polymerized dendritic compounds are nowadays an established polymer category next to their linear, branched, and cross-linked counterparts. Their uncommon tree-like architecture is characterized by adjustable internal cavities and external groups. They are therefore exceptional absorbents and this attainment of high concentrations in their interior renders them ideal reaction media. In this framework, they are applied in many environmentally benign implementations. One of the most important among them is water purification through pollutant decomposition. Simple and composite catalysts and photo-catalysts containing dendritic polymers and applied in water remediation will be discussed jointly with some unconventional solutions and prospects.

## 1. Introduction

The field of nanoscience has witnessed particular research attention and growth in the last two decades, thanks to its vast applicability in several areas, including catalysis, optoelectronics, sensing, drug delivery, energy storage, and water decontamination [1,2,3,4,5,6,7,8]. Water pollution is a serious global environmental phenomenon that poses a major threat for both the maintenance of the aquatic ecosystem and human health and welfare. Among all different water pollution sources, the industrial sector takes the lead, producing the highest amount of organic and inorganic wastewaters, most of which are toxic and non-biodegradable and can produce carcinogenic or mutagenic effects in tissues of living organisms through ingestion, inhalation, and skin contact [9,10,11]. Specifically, industrial phenolic by-products, dyes, and pigments constitute some representative examples of such water contaminants, whose increased solubility in water, along with their high resistance to common degradation practices, make their removal from water environments a rather complex, costly, and time-consuming process. Some of the most popular techniques for eradicating these color substances from water are adsorption [12], photocatalysis [13], membrane technology [14], chemical oxidation [15], biological degradation, and coagulation [16,17]. The first and second categories are the most employed methods due to their operational simplicity and non-toxic nature. However, the limited recyclability, low stability, and high solubility of the adsorbent/catalyst decrease their water remediation performance and necessitate the development of novel, more efficient methods for water treatment purposes [18].

The introduction of dendritic molecules revolutionized the field of nanoscale materials and polymer technology. Representing a separate class after their linear, branched, and cross-linked counterparts they constitute the most extensively utilized type for the fabrication of nanoscale composites with adjustable architecture and controlled physicochemical properties [19,20,21,22,23]. Dendritic architectures can be classified into five main categories based on their special structural and dispersity characteristics: (a) monodispersed symmetric dendrimers, (b) asymmetric hyperbranched polymers, (c) dendrons, fragments of dendrimers and hyperbranched polymers emanating from a focal point, (d) dendronized polymers, deriving by chemical binding of dendrons to linear polymers, and (e) dendrigrafts, a mixed category of hyperbranched polymers with repeated units of linear polymers and branched architecture [24,25,26,27,28] (Figure 1). Interestingly, the characteristic branched tree-like morphology of these macromolecules, inspired the choice of their name, which comes after the Greek word δένδρο (i.e., tree) [29]. Dendritic polymers among a multitude of potential applications are considered novel candidates for use in water remediation treatments [14,30,31,32,33]. They can be synthesized through two main mechanisms: the divergent and the convergent [34,35,36]. In the divergent method, dendrimers’ synthesis initiates from a focal point of a reactive core, whereas in the convergent method, independently synthesized dendrons are assembled to produce the final dendritic structure [37,38,39] (Figure 2a,b).

The predominant advantage of dendritic materials is the hosting environment that they offer, which originates from the completely manageable synthetical procedure of their building blocks (e.g., functional groups of the interior branches and the exterior surface) [40]. The fabrication of hybrid nanoparticles with desired shapes and physicochemical and biological functionalities [25,41,42,43,44,45] is thus facilitated. More particularly, tailored dendritic-metal nanocomposites can be produced, when guest metal ions can either be encapsulated in the internal “branches” of dendrimers or be stabilized with their peripheral functional end-groups. In the same way semiconductors, and other active substances of nanosized dimensions can be incorporated that combine the properties of both the dendrimer and the hosting material [27,46]. Some of the most known and widely used dendrimers in the field of hosting technology are diamino butane poly(propylene imine) DAB/PPI and poly(amidoamine) PAMAM. The most common non-symmetric hyperbranched counterparts are poly(ethylene glycol), polyglycerol PG, and poly(ethylene imine) PEI [47,48] (Figure 3). Dendrimers as individual entities may be employed in a multitude of applications. Interestingly, their ability to act as biomimetic materials attracted many medicinal fields to use them for applied research purposes as drug delivery carriers, diagnostical and imaging tools, anticancer, and gene therapy [49,50,51,52]. Additionally, fine-tuning of their guest-host interrelations favors their use in separation technology as homogeneous catalysts and interestingly as inexpensive, water-insoluble adsorbents of organic and inorganic water contaminants [53,54,55,56,57,58].

Transition metal nanoparticles are great candidates for use in catalytic reactions while their oxides and sulfides are suitable for photocatalysis. Owning to their special electromagnetic properties, electroconductivity, and wide reactive surface area, which promotes the interfacial interaction between the active substrate and the reactants, these nano-catalysts turned out to be particularly useful in the degradation processes of organic contaminants for water purification. However, their increased susceptibility towards agglomeration and oxidation, both of which significantly downgrade their catalytic potential, propelled the discovery of novel matrices and protective stabilizers that prevent the occurrence of such phenomena during catalytic procedures [59,60]. Among the various encapsulating nano-agents that can be employed as passivators, including carbon, clay, silica, zeolites, and organic polymers, the latter are the most established and widely used ones. Specifically, thanks to their ability to envelop molecules, dendritic polymers have been proved to be ideal supports for the fabrication of colloidal solutions imparting higher chemical stability, increased solubility, and thus enhanced catalytic properties [6]. The grafting of dendrimers on soluble nanoparticles surfaces is markedly advantageous since it enables the control of the size and site of the interfacial reaction, [61,62,63]. Current literature proves the successful formation of several metal nanoparticles, such as Pt, Pd, Ag, Cu, Au, Ru, and Rh, encapsulated in dendrimers by following the colloid template synthetical method [64,65]. Bimetallic nanocomposites contained in dendrimers can also be achieved through three different reaction methods: partial displacement, co-complexation, and the sequential loading method. Semiconductor oxides, sulfites are also governed by the same nucleation concepts. These nano-catalysts with dendrimers as templates have been repeatedly utilized in the past for numerous catalytic and photocatalytic reactions, including oxidations, reductions, hydrogenations, and covalent bond cleavages that all aim at the biodegradation of toxic wastewater substances, avoidance of secondary wastes formation, and recyclability of materials used for targeted water purification applications [12,13,16,17,66].

The present review will characterize in great detail the methods used for the synthesis of catalysts that are based on nanoparticles encapsulated in dendrimers and compare their efficacy specifically in water decontamination processes. Some special cases involving dendritic polymers but not necessarily anticipated guests (e.g., ions or enzymes) will be presented as well. Prospects of these materials will also be included in an attempt to optimize their properties and expand their applicability in other pollutant categories.

## 2. Conventional Homogenous Catalysis by Metal Nanoparticles

### 2.1. Reduction of Aromatic Nitro-Compounds

#### 2.1.1. The Benchmark Reaction of *p*-Nitrophenol

The reduction of nitro compounds to their amino counterparts, in general, transforms highly toxic compounds into less toxic analogs that may be recovered and are usually useful in pharmaceutical antipyretic and analgesic preparations. Specifically, *p*-nitrophenol is mutagenic [67] cyto- and embryotoxic [68] and for this reason, it constitutes one of the most burdensome compounds in industrial and agricultural wastewater. This explains the fact that it is the most common standard used for the initial catalytic evaluation of metal NPs formed by dendritic polymers. Furthermore, it provides many insights about basic nucleation principles and the overall NPs properties that affect pollutant transformation processes.

The earliest attempt to exploit the potential of these dendritic polymer-metal nanocomposites to *p*-nitrophenol hydrogenation was implemented in 2003 by Hayakawa et al. [69]. The group introduced laser irradiation for the production of gold NPs with PAMAM G3, G5, and PPI G3, G4. They observed a general tendency of the NPs average diameters to decrease by dendrimer concentration and most importantly suggested that the dendritic macromolecules form monolayers around the metal cores. Esumi expanded the scope to silver, platinum, and palladium [70] and Bingwa performed kinetic analysis and the determination of the Langmuir–Hinshelwood parameters for the particular p-nitrophenol reduction catalyzed by Ag and Au NPs [71]. In all these works, NP size was independent of dendrimer generation and PPI nanocomposites exhibited the best reduction rates. The interest for this novelty was so high that as soon as 2009, 4-nitrophenol reduction by PAMAM-Cu combined NPs was proposed as a laboratory experiment suitable for first-year university students [72].

In contrast, to the previous experiments in the case of Pd incorporation to PAMAM G4 and PAMAM-OH G4, G6 cavities Johnson et al. reported the synthesis of agglomerates with numbers of atoms approximately equivalent to the dendrimer/Pd^2+^ ratio. The authors correlated their activity to their size indicating that for clusters up to 50 Pd atoms there is a positive linear dependency to the reaction rate. They also established the steric hindrance of the larger generation dendrimers to the accessibility to the catalysts [73] which was confirmed by Nemanashi et al. for Ag, Cu, and Au NPs [74]. PAMAM-OH G6 though exhibited lower activation enthalpy (H‡) [75] activation energy (EA) [75,76] and effectively inhibited reoxidation of Cu^0^ NPs to Cu^2+^ [77]. A direct comparison of Pt and Pd NPs formed by sodium diethyl hexyl sulfosuccinate reverse microemulsions in an isooctane/water system with their respective counterparts templated by PAMAM-OH G4-G6 revealed that the latter have about half average diameters, narrower dispersity, and superior catalytic performance and established the potential of the dendritic scaffolds method [78].

The first silver core/gold shell alloy particles with sizes 3–4 nm were obtained by sequential incorporation of Ag^+^ and AuCl_4_^−^ ions into the cavities of amine-terminated G3, G5 and carboxyl-terminated G3.5 5.5 half generations poly(amidoamine) dendrimers [79]. Hydroxyl functionalized G6 additionally stabilized Au, Cu, Pd, Pt, and Au-Cu, Pd-Cu, Pt-Cu bimetallic NPs with smaller diameters in the range of 1 to 3 nanometers and established the beneficial synergistic effect of Ag or Cu combination with a more expensive noble metal to the reaction rate [80]. The analogous outcome was created when PAMAM-OH G4-G6 were employed to generate Ru NPs with mean diameters ranging from 1.1 to 2.2 nm [81] and to amalgamate Ru and Ni with the bimetallic combination manifesting superior performance that single metal equivalents [82].

The 4-nitrophenol hydrogenation process model was followed further to evaluate most of the newly synthesized dendritic matrices. These are being designed in order to compose task-specific metal nano-catalysts. 1,2,3-Triazolyl click dendrimers containing silicon and external pentose groups (Figure 4a) were conceived to secure stoichiometric coordination of Au [83] and Pd [84] ions to the internal triazole groups. This set ratio led to the nucleation of smaller metal cores and larger overall active surfaces that were also able to catalyze Suzuki–Miyaura coupling of 4-bromoacetophenone with phenylboronic acid. Replacement of the pentose groups by nona-PEG-chains (Figure 4b) afforded gold NPs of diameters 1.8 and 12 nm depending directly on the HAuCl_4_/star polymer analogy. Furthermore, longer PEG chains (PEG2000 instead of PEG550) favor smaller metal particles [85]. Further alternative substitution by nine arene cores and 27 triethylene glycol chains (Figure 4c) led to a matrix capable to agglomerate a wide variety of transition metals (Fe, Ru, Co, Rh, Ir, Ni, Pd, Pt, Cu, Ag, and Au) from simple commercially available salts. All reaction rates were slower in comparison to analogous composites stabilized by simple triazoles due to the stereochemical restrictions of *p*-nitrophenol diffusion through the bulky periphery of the nano-catalyst. Pd Nps were the smallest, the most effective, and did not require induction time [86].

Interaction of four different generations (G1-G4) of similar triazolyl glycodendrimers (Figure 4d) with gold cations revealed strong evidence towards the support of the theory of two nucleation mechanisms first noticed by Tanaka for Pd [87] and further described by Noh [75] (Figure 5a). Stabilization of metal NPs into an organic cluster was observed for the two smallest dendrimers while in the case of the larger analogs incorporation occurred into their cavities [88]. Related amphiphilic Janus dendrimers bearing benzotriazole units and hydrophilic oligo (ethylene glycol) (OEG) chain on one side and a hydrophobic benzyl ether dendron functionalized by two long aliphatic chains formed micelles as well and incorporated Ag and Cu NPs [89] (Figure 5b). In a final most recent variant a 1,2,3-triazolyl dendronized polymer formed micellar catalytic nanoreactors when decorated by Au NPs (Figure 5c) [90] Random copolymerization [91] and controlled di-block polymer formation [92] with ferrocene-containing dendrons, by ring-opening metathesis, afforded mixed dendronized polymers that self-assemble into spherical micelles (Figure 5d,e), are capable of Ag and Au NP stabilization and the composites may perform 4-nitrophenol reduction with an extremely high turnover number up to 25,000.

**Figure 4 nanomaterials-12-00445-f004:**
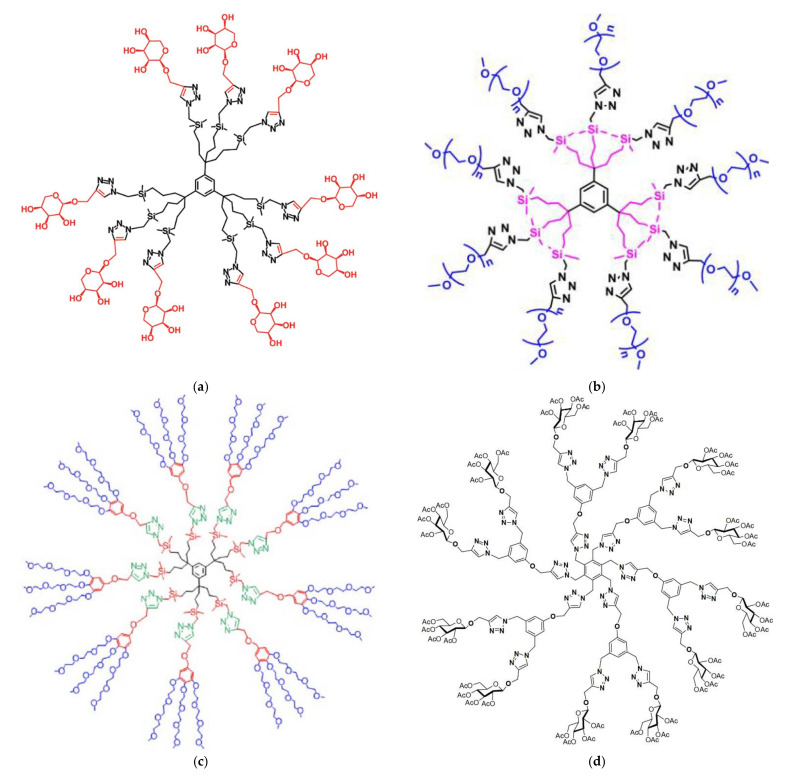
(**a**) Triazolyl Pentose-Terminated click dendrimer. Reproduced with permission from [83]; (**b**)Triazolyl nona-PEG click dendrimer. Reproduced with permission from [85] (**c**) Triethylene glycol nona-arene triazolyl click dendrimer. Reproduced with permission from [86] (**d**) Triazolyl glycodendrimers. Reproduced with permission from [88].

**Figure 5 nanomaterials-12-00445-f005:**
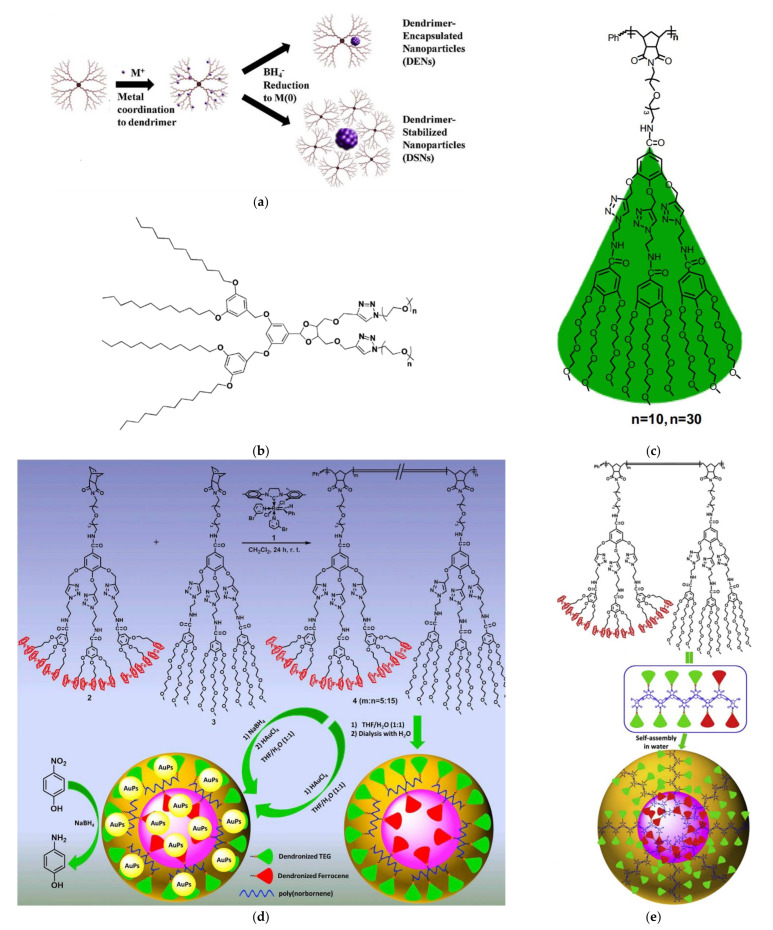
(**a**) Schematic illustration of the two dendrimer templated metal NPs nucleation mechanisms. Reproduced with permission from [75]; (**b**) Chemical structure of a dendritic Janus amphiphile benzotriazole. Reproduced with permission from [89]; (**c**) 1,2,3-triazolyl dendronized polymer. Provided by the Springer Nature SharedIt content-sharing initiative from [90]. (**d**) Synthesis and self-assembly of mixed dendronized triethylene glycol-ferrocene random copolymer. Reproduced with permission from [91] and (**e**) diblock copolymer. Reproduced with permission from [92].

Among other efforts worth mentioning, Asharani and Thirumalai reported the synthesis of PEG-G1-(3,5-DHB-OH)16 (Figure 6a) a poly(ethylene glycol core) dendrimer bearing hydroxyl terminal groups and paired it with silver cations [93]. Analogous successful catalytic Ag NPs combination was implemented with polyetheramine (Jeffamine) core PAMAM dendrimer [94]. Functionalization of PPI G2 by 1,2-epoxyhexane afforded an amphiphilic template (Figure 6b) for the aggregation of extremely small-sized Ru NPs in the range between 0.5 to 1 nm [95]. Similar core-shell architecture was produced by the hyperbranched analog PEI as the core and linear or dendritic hydrophobic shells. These copolymers were designed to increase the stability of AuNPs of average diameters of 3 nm (Figure 7a). The branched architecture of the latter offered higher catalytic rates and an “unprecedented” maximal turnover of approximately 23,000. [96] When ammonium salts of hyperbranched polystyrene were employed even smaller gold nanoparticles (1 nm in size) were achieved and were related to the observed “unique” reduction rates. Apart from the conventional *p*-nitrophenol transformation testing, aerobic oxidation of alcohols and homocoupling of phenylboronic acid were successfully attained [97]. Another series of amphiphilic (PEI) core and acetic amide, propionic amide, butyric amide, and isobutyric amide shells containing Au NPs established the superiority of the branched shells as optimal catalytic activity and reaction rates were achieved by the isobutyric amide functionalized derivative. The linear shells followed the rule of stereochemical diffusion limitations with the less bulky acetic amide/PEI/Au NPs nano-catalyst exhibiting the second-best results [98]. The most recent homogenous reduction catalysis implementation is by far the most unconventional. A mechanochemical method of blending solid powders of metal precursors [Ir(cyclooctadiene)Cl]_2_ and PtCl_2_, with triazine dendritic hosts (Figure 7b) and NaBH_4_ milling and then extracting the water-soluble impurities from an H_2_O/CH_2_Cl_2_ biphasic system was proposed for obtaining Ir and Pt NPs [99]. In order to enhance the biocompatibility of PAMAM G5/Au NPs coupling of the dendrimer with maleic anhydride and then with cysteamine was performed. Reduction of the metal ions took place in a solution of chloroauric acid sodium borohydride. The resulting materials retain the catalytic properties of the unmodified parent compounds. Furthermore, due to a zwitterionic layer consisting of amino and carboxyl groups they formed aggregates neither in fibrinogen nor in phosphate-buffered saline solutions and they were also far less toxic [100]. In the same framework attempt to reduce PAMAM toxicity by coupling with 4-carbomethoxypyrrolidone resulted in biocompatible scaffolds that hosted smaller than 5 nm silver NPs and exhibited improved performance as well [101].

It was also discovered that dendritic composites containing PEI moieties not only stabilize AuNPs, but they may also act as reducing agents. The first such example was the outcome of the synthesis of a tripartite dendritic copolymer containing commercial aliphatic hyperbranched polyesters (Boltorn Hx) as cores and alternating linear poly (ethylene glycol) PEG chains and hyperbranched PEI at the periphery (Figure 8) [103]. By starting from 2,2-bis(methylol)propionic acid polyester (H104) core and attaching polyethylene glycol monomethyl ether chains and a variety oligo(ethylenimines) a resembling hybrid amphiphile is produced and induces the formation of Au NPs with diameters 2–4 nm (Figure 9). In this case, the branched triaminoethyl amine and the longest linear tetraethylenepentamine present the best catalytic activity and the most efficient protection. The beneficial stabilizing effect of the multiple amino groups profoundly overcomes the steric effects [104]. A third dendronized copolymer containing three moieties as well (PEG-PEI-PCL) was produced for stabilizing Pt NPs by attaching hyperbranched PEI to one linear poly (ethylene glycol) chain and then grafting poly(ε-caprolactone) chains to the terminal amino groups. Self-assembly in aqueous solutions afforded organized micelles producing upon reduction of PtCl_6_^2−^ large nanoparticles (8 nm) suitable for catalytic hydrogenation of 4-nitrophenol [105] All the homogenous catalysis systems researched and discussed herein for the specific nitrophenol reaction are summarized in Table 1.

#### 2.1.2. Aromatic Nitro-Derivates in General

As expected, inspiration from the first successful experimentations with 4-nitrophenol drove for further investigations with close related nitroaromatic pollutants. The first such expansion was carried out by Yang et al. in 2006. Polyaryl ether trisacetic acid ammonium chloride dendrons were used for the reduction of platinum ion and subsequent organization around the resulting Pt NPs (Figure 10a). The mean diameters of the latter ranged from 2.0 to 5.5 nm, depending on the metal to dendron molar ratio. The catalysts were able to accomplish hydrogenation of *p*-nitrophenol, *o*-nitroanisole, *o-,m-,p*-nitrotoluene to *p*-nitroaniline, *o*-anisidine and *o-,m-,p*-aminotoluene, respectively, under an atmospheric pressure of H_2_. Activity dropped by increasing dendron generation [106]. The same structures were used for the assembly of Au/Pt bimetallic core/shell NPs with 6-nm Au cores and average overall sizes of 9.0 ± 2.4 nm, 10.4 ± 2.8 nm, and 13.0 ± 3.2 nm as a function of Pt/Au molar ratio and improved performance over monometallic platinum. Apart from the already discussed variety of nitrobenzenes, the catalyst was successfully tested for the catalytic hydrogenation of 3-phenoxybenzaldehyde which comes from the environmental transformation of the pesticide β-cypermethrin [107]. Au NPs reduced and stabilized by amphiphilic PPI were effective for nitrobenzene conversion to aniline as well [108].

Besides the amino groups of PPI dendrimer and hyperbranched PEI hydroxyl functionalities of dendritic PEG may also act as reducing agents for metal cations. The first such example is reported lately by Asharani. Polyethylene glycol (PEG) core dendrimer reduced encapsulated silver ions to Ag NPs and effectively catalyzed a wide range of substituted nitroaromatics under the following reduction rate order 4-nitrobenzaldehyde > nitrobenzene > 4-nitrocatechol > 4-nitroaniline > 5-hydroxy-2-nitrobenzaldehyde > 4-nitro anisole > 2-hydroxy-5-nitrobenzylbromide [109]. Functionalization of PAMAM G3 terminal amino groups with long linear PEG chains also permitted the assembly of Cu [110] (Figure 10b) and Au NPs [111] (Figure 10c). Reduction of 4-(4-nitrophenyl) morpholine and two pharmaceutically important derivatives (4-(2-fluoro-4-nitrophenyl) morpholine) and 4-(4-nitrophenyl) morpholin-3-one was attained. The best performance was observed at a 0.4 PAMAM/Metal ratio. Hydrogenation of nitroaromatics was even reproduced by Cu NPs incubated into simple PAMAM G2 dendrimers by hydrazine hydrate instead of NaBH_4_. Nine modified aromatic nitro benzenes were tested (4-nitrophenol, 2-nitrophenol, 4-nitrobenzaldehyde, 2,4 dinitrophenol, 2-nitroaniline, 4-nitroaniline, 3-nitrotoluene, 4-nitrotoluene, 4-nitrochlorobenzene). Chemoselective reduction of nitro groups in the presence of other reducible groups such as aldehydes was detected as previously and even regioselective reduction of 2-nitro group close to the hydroxyl in the case 2,4 dinitrophenol was recorded [112]. The catalytic composites presented herein for nitro-derivative reduction are summarized in Table 2.

**Figure 10 nanomaterials-12-00445-f010:**
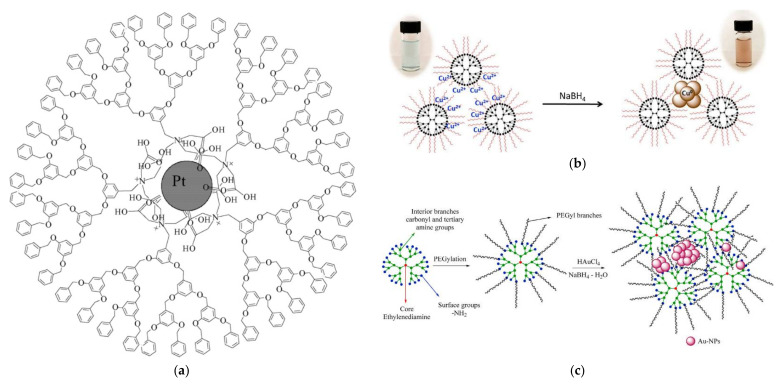
(**a**) Organization of Polyaryl ether trisacetic acid ammonium chloride dendrons around Pt NPs. Reproduced with permission from [106]; (**b**) Incorporation of Cu NPs into the dendrimer layer of PEG-PAMAM G3. Reproduced with permission from [110]; (**c**) Synthesis of PEG-PAMAM G3 Au NPs. Reproduced with permission from [111].

### 2.2. Treatment of Pigments

The next step in research for catalytic handling of water pollution originates from problems caused mainly by the textile industry and other sources of pigment and dye effluents. Considering the early successes of the preliminary efforts with metal nanoparticles enclosed into dendritic polymers the evolution in this field is rather delayed. An initial example of such investigation is the employment of quaternary octyl ammonium salts of PPI dendrimer for the induction of Ag, Pd, Pt NPs. Amphiphile catalysts with 5 and 10 alkyl chains were used for the reduction of methyl orange to 4-aminobenzenesulfonic acid and 4-dimethylamino aniline. Palladium proved more suitable over silver and platinum and as expected from the stereochemical point of view the longer alkyl chains slowed the diffusion of the bulky pollutant [113]. The formation of bimetallic Pd/Au nanoparticles into PAMAM-OH G6 produced an additional successful suggestion for methyl orange degradation that established the excellent potential of Pd NPs [114].

Conventional PAMAM G5 with Au, or Ag NPs [115], PAMAM G6 Au hybrids [116], and hydroxy-terminated PAMAM-OH G6 dendrimers combined with monometallic Pd and Pt NPs [117] or bimetallic Pd/Au alloys [118] were employed for the oxidative degradation of morin by hydrogen peroxide to morin oxide. In addition, Pd NPs stabilized into the previously described biocompatible system i.e., G5 PAMAM functionalized with maleic anhydride and cysteamine, were also checked for their ability to perform this particular reaction in bacteria-contaminated solutions [119,120]. Diffusion of the reactants through the dendritic periphery and subsequent adsorption on the metal NPs surface was again the critical step defining the reaction constants. The presence of palladium proved particularly beneficial for this oxidative path as well, leading to lower activation energies and faster reaction rates.

In another instance, methylene blue underwent catalytic oxidative decomposition by the mediation of PAMAM G5 and Au or Ag NPs [121]. The same dye is treated by an alternative reductive path by sodium borohydride. Interestingly among others (PAMAM-OH G4, G5 with Pd NPs, PAMAM G4 with Au) the same catalytic composite used for oxidation PAMAM G5/Au may also accomplish the reduction. Once more Pd NPs were the most efficient catalyst [122]. Examples of catalytic degradation paths for methyl orange, morin, and methylene blue by metal NPs incorporated into PAMAM are depicted in Figure 11. A summary of all the formulations employed for homogenous dye discoloration is presented in Table 3.

## 3. Formulations and Solid Supports for Heterogeneous Catalysis

### 3.1. Early Evolution Efforts by Following the p-Nitrophenol Standard

Grafting the dendritic polymer on the surface of organic or inorganic support was the obvious choice to secure the easy recovery of the catalyst. The first such substrate was crosslinked poly(4-vinyl pyridine) beads for PPI G2 dendrimer bearing Au NPs. The catalyst was removed by simple filtration and could be reused at least 10 times Due to its spherical shape it can be applied as filling material in continuous column reactors [123]. Polystyrene microspheres bearing carboxy reactive groups present one more spherical organic base suitable for divergent PAMAM G5 dendron expansion. After the typical incorporation of silver NPs, a hierarchical structure is formed by coating with silica shell via an improved Stöber method to construct a different packing medium for stainless steel column reactors [124]. A further very popular organic substrate cellulose in the form of nanocrystals can be oxidized via (2,2,6,6-tetramethyl piperidin-1-yl)oxidanyl, TEMPO regent to form surface carboxy groups. Attaching PAMAM G6 through carbodiimide-mediated amidation reaction allowed two different Au NPs nucleation mechanisms (Figure 12a). Encapsulation of NPs having diameters of approximately 2 to 4 nm at neutral pH with NaBH_4_ and stabilization of NPs with sizes 10 to 50 nm at pH 3.3 with no additional reducing agent. As usual, the smaller NPs are the more effective [125].

Polymer fibers present also an interesting substrate as they may form dendronized polymers by grafting with dendritic polymers. One such example is the grafting of hyperbranched PEI to polyacrylamide through hydrolysis and amidation. The copolymer produced small Au NPs around 3 nm. Particle size increased stereotypically by decreasing dendritic polymer generation. This configuration presented an extremely high turnover value of 50,000 [126]. Another two strategies practiced in a related substrate consisted from electrospun polyacrylic acid (PAA)/polyvinylalcohol (PVA) nanofibrous mats was to bind low generation PAMAM G2 either through supramolecular electrostatic interactions or through the covalent 1-ethyl-3-(3-dimethylaminopropyl) carbodiimide hydrochloride (EDC) reaction. Surprisingly after doping with 5 nm Au NPs the physically coupled polymer exhibited the best performance without undesired side effects such as polymer leaching [127].

As expected, the hybrid inorganic-organic-metal nanoparticle composite catalysts were initially based on ceramic silica and specifically ordered mesoporous SBA-15 SiO_2_. The reaction of the ceramic with aminopropyl triethoxysilane (APTES) functionalized its surface with reactive amino groups. Conventional divergent PAMAM G4 dendron propagation by alternating Michael additions of methyl acrylate and amidations with ethylenediamine and reduction of Pt^4+^ afforded 1.2 to 2.6 nm Pt NPs (Figure 12b). Apart from 4-nitrophenol, the catalyst was able to convert ferricyanide to ferrocyanide by thiosulfates [128]. Silica in the form of glass microreactors reacts with APTES as well. Subsequent interaction by *p*-phenylene diisothiocyanate covalently attached PAMAM G4 to their inner walls. Co-complexation of AgNO_3_ and HAuCl_4_ in the dendritic template resulted in bimetallic NPs. Catalytic performance was heavily dependent on Ag/Au stoichiometry with the best results obtained at a 1:1 ratio [129].

Magnetic supports originating predominantly from a Fe_3_O_4_ core, offer even more easy separation by the simple application of a magnet. In an early implementation after coating the magnetic nucleus with a preparatory polystyrene layer, PAMAM G7 dendrons emanated and incubated the formation of Ag NPs [130]. From the same magnetite core reaction with APTES introduced a silica interlayer that permitted posterior covering with PAMAM G1 via the typical divergent reaction to the reactive amino groups (Figure 12c) and encapsulation of narrowly dispersed Ag NPs to give another variant of magnetically recyclable catalyst [131]. More sophisticated conformations were equally developed. Successive coatings first with 3-methacryloxypropyltrimethoxysilane KH-570 silane coupling agent, then with glycidyl methacrylate and divinylbenzene by distillation precipitation copolymerization and finally with hexamethylenediamine to pin primary amine groups pending from the microspheres through a six-carbon spacer provided an ideal substrate for PAMAM propagation without stereochemical limitations (Figure 13a). Coupling with Au Nps led to a composition capable of achieving up to 78.7% 4-nitrophenol degradation after ten cycles [132] A closing reference to inorganic magnetic substrates will be made to an exceptional idea completely “out of the box” that concerns dendritic architecture based exclusively on silica. After an initial sol-gel coating of the magnetic core with tetraethoxysilane (TEOS), a dendritic SiO_2_ shell is formed by the oil-water biphase stratification method. Decoration of these branches with external amino groups by treatment with APTES and doping with silver NPs generated a catalyst suitable for 4-nitrophenol and 2-nitroaniline [133].

**Figure 12 nanomaterials-12-00445-f012:**
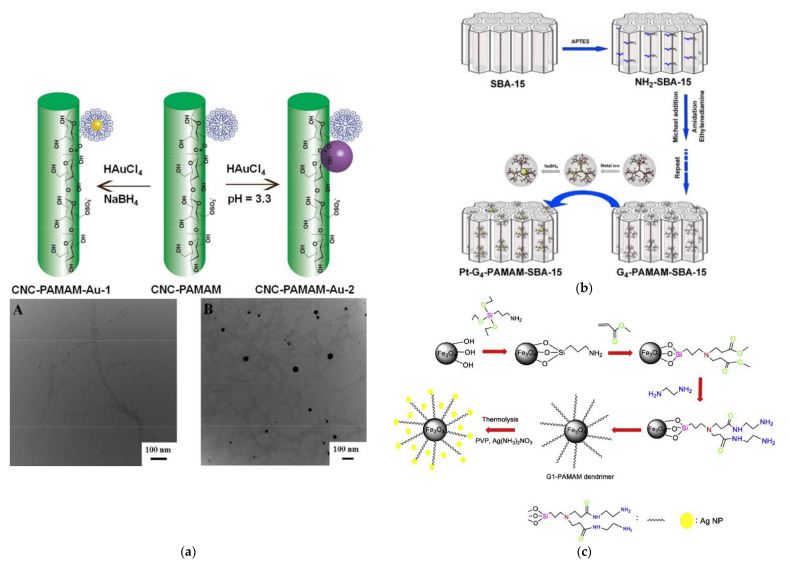
(**a**) Schematic illustration of the synthetic process and TEM images (A and B) for Au NPs formed by two distinctive nucleation mechanisms. Reproduced with permission from [125]; (**b**) Procedure for the synthesis of SBA15-PAMAM G4-Pt Reproduced with permission from [128]; (**c**) Schematic illustration of the formation of Fe_3_O_4_@APTES@PAMAM G1-Ag Magnetically Recyclable Catalyst Reproduced with permission from [131].

Carbon allotropic forms constitute further promising surfaces for dendritic polymer grafting [134]. In a first combination, hyperbranched PAMAM dendrons emanate from graphite surface activated by KMnO_4_ oxidation, (Figure 13b). Leaving aside the ordinary p-nitrophenol, embodied Ag NPs enabled the reductive transformation of a wide variety of nitroaromatics (nitrobenzene, *o*-nitrophenol, *o*- and *p*-nitroaniline, *p*-nitrotoluene, (*p*-nitrophenyl)methanol, 4-chloro-1,2-dinitrobenzene, *o*-chloronitrobenzene, *p*-bromonitrobenzene, *p*-iodonitrobenzene, (1*E*,1′*E*)-N,N′-(cyclohexane-1,2-diyl)bis(1-(4-nitrophenyl) methanimine)). No side dehalogenations or hydrogenations of the imine groups occurred indicating a high level of chemo-selectivity [135]. In a synergistic scheme, magnetic graphene oxide is obtained through doping with MnFe_2_O_4_ NPs by solvothermal technique. Following typical induction of amino groups by APTES and PAMAM G3 dendron evolution; the immobilization of Pd NPs (Figure 14) led to a catalyst with reducing potential towards an extensive range of nitropollutants: 2-nitroaniline, 2-chloro-6-nitrophenol, 1-chloro-4-nitrobenzene, 1,2-dinitrobenzene, and 2,5-dinitrophenylhydrazine [136]. Multi-walled carbon nanotubes (MWCNTs) may be activated by similar induction of carboxyl groups onto their surface by oxidative procedures such as KMnO_4_ phase transfer catalysis. Chemical attachment of PPI G2, G3, dodecyl quaternary ammonium groups and immobilization of Pd and Ag NPs produced a recyclable, five-fold more efficient catalytic agent in comparison to the simple metal dendritic polymer hybrid [137,138]. Moreover, this composition proved a very efficient antibacterial agent for the Gram-negative representative pathogen *Pseudomonas aeruginosa* and the respective Gram-positive *Staphylococcus aureus*. [139].

**Figure 13 nanomaterials-12-00445-f013:**
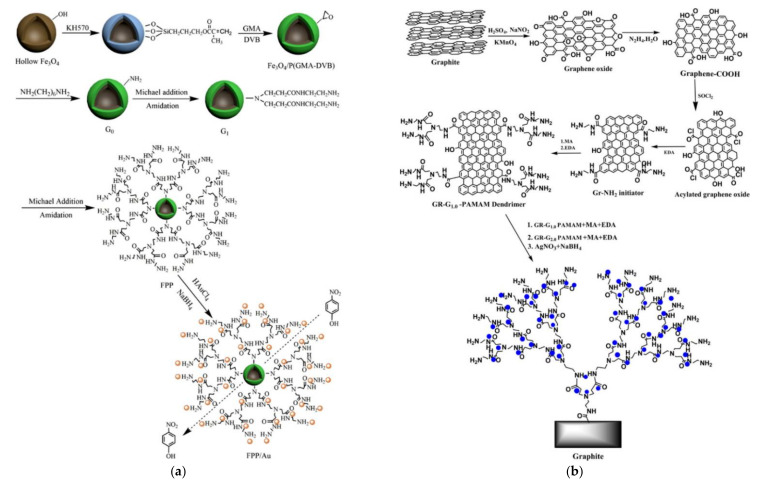
(**a**) Schematic representation of the process for the synthesis of magnetic Fe_3_O_4_/P(GMA-DVB)/PAMAM/Au microspheres Reproduced with permission from [132]; (**b**) Procedure for the synthesis of graphite-PAMAM G3-Ag dendrimer. Reproduced with permission from [135].

**Figure 14 nanomaterials-12-00445-f014:**
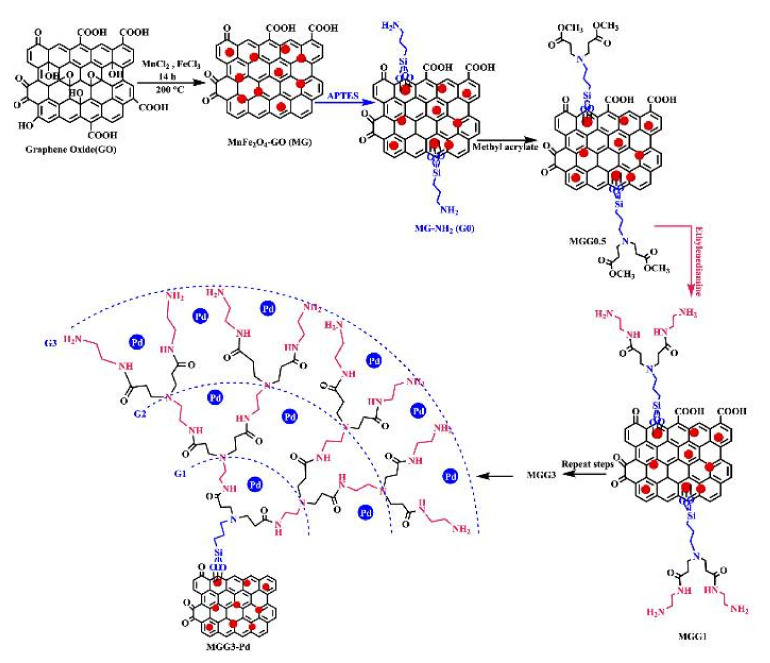
The synthetic procedure of the graphene oxide-MnFe_2_O_4_-PAMAM G3-Pd nanocomposites. Reproduced with permission from [136].

A very simple concept defining the limits of homogenous catalysis and where the realm of heterogeneous mechanisms begins concerns the use of a dialysis membrane to enclose a PAMAM G4 Au NPs solution prepared as usual. Immersion of this container to a solution containing p-Nitrophenol and NaBH_4_ initiates the common catalytic reaction. After the reaction termination, the catalytic solution is easily isolated and may be reused after recycling in a new catalytic cycle [140]. In a second fundamental idea amphiphilic dendritic polymers consisting of a PEI core bearing hydrophobic shells of saturated palmitamides, octadecanamides, or unsaturated oleamides enhanced the stability of spherical Au NPs and their solubility in organic solvents such as petroleum ether, n-butyl acetate, and toluene enabling them to be employed in heterogenous biphasic catalytic reduction systems with turnover number values up to 5040 [141].

PEI core combined with a poly (styrene-co-2-ethylhexyl acrylate shell was combined with Pt nanoclusters (Figure 15a) [142,143] whereas a slightly different shell containing polystyrene and dodecyl aliphatic chains was paired with Au NPS (Figure 15b) [144]. The reaction of the intermediate composite in a biphasic water-toluene system caused organization into high internal phase emulsion. Radical emulsion polymerization initiated by azodiisobutyronitrile produced open-cellular, elastic monoliths (Figure 15c,d) that proved highly recyclable catalysts with excellent properties. The heterogeneous systems designed and tested for nitro-derivatives elimination are listed in Table 4.

### 3.2. Heterogeneous Catalytic Degradation of Dyes

In a second implementation step, aligned with homogenous catalysis, the experience gained from the interactions of heterogenous dendritic polymer-NP hybrids with nitrophenol and the overall family of nitroaromatic compounds was put into action for dye decomposition. Research on this particular subsector is quite fresh covering only the last seven years. Covalent grafting of PAMAM on graphene oxide as previously described provided a scaffold for the formation of bimetallic Ag/Au NPs. The products were able to eliminate commercial colorants used in the dyeing industry such as methyl orange and Congo red [145]. An impressively interesting proposition introduces dendrigrafts to water purification technologies and integrates benefits from two different classes of dendritic polymers with distinct functionalities. Notably, 6-armed PEG-NH_2_ covalently interacts with the epoxide, hydroxyl, and carboxyl oxygenated functions of nanographene oxide (PEGylation) and then thiol-functionalization occurs without (Figure 16a) or with intermediate amidation reaction with PAMAM (Figure 16b). Au NPs evolved from the thiol sites, are effectively reducing 4-nitrophenol, 4-nitroaniline, and Congo red [146].

A second inorganic point of view consists in the physical immobilization of PAMAM-OH G5 dendrimer encapsulating random alloy Pd-Au NPs (2.66 ± 0.51 nm) on reducible mesoporous transition metal oxides (CeO_2_, NiO, Co_3_O_4_, Fe_2_O_3_, MnO_2_, and SiO_2_) by sonication. The synergistic catalytic activity of the ceramic host [147] with the metal guest was established. The best results in morin oxidation were demonstrated by Pd/Au–Co_3_O_4_ and Pd/Au–CeO_2_ [148]. This synergy was verified when platinum NPs of different sizes (Pt_n_, n = 55, 140, 225) nucleated into PAMAM G6-OH were adsorbed via wet impregnation procedure into mesoporous Co_3_O_4_ synthesized by inverse micelle method. Oxidation of methylene blue proceeded faster by the mediation of these composites in comparison to PAMAM G6-OH-Pt adsorbed in simple silica and kept up for at least 8 cycles with minimum metal leaching [149].

Organic substrates play a protagonistic role in dendritic polymer grafting for composite catalyst manufacturing. As was the case with GO, experiments with organic polymer fibers and 4-nitrophenol promoted the development of dispersions for dye effluents treatment. In a first approach 2,2,6,6-tetramethylpiperidine-1-oxyl radical-oxidized nanofibrillated cellulose. Association with PAMAM G4 and AgNO_3_ produced an effective multifunctional material for reductive discoloration of Rhodamine B and antibacterial action against both Gram-positive and Gram-negative bacteria [150]. Polyester (PET) fibers functionalized by PAMAM or APTES and decorated by Cu and Ag NPs were tested at first for their reducing potential towards 4-nitrophenol, methylene blue, malachite green, and Remazol red. In a second stage, multifunctional filters based on a three-layered sandwich structure PET1-PAMAM|PET2-APTES-Ag/Cu|PET1-PAMAM of nonwoven fabrics were manufactured from these fibers. They accomplished simultaneous dissolved solids filtration, bacteria disinfection (*Escherichia coli*, *Staphylococcus epidermis*), and catalytic p-nitrophenol and dye discoloration in continuous flow [151]. Air atmospheric plasma treatment to hydrophobic PET fibers of a nonwoven membrane introduces hydrophilic carboxy and hydroxy groups that facilitate the optional step of hyperbranched poly-(ethylene glycol)-pseudo generation 5 chemical binding. Ferric ions adsorbed to these functionalized substrates are transformed to zero-valent iron (ZVI) via NaBH_4_ by two different mechanisms either one step (in-situ simultaneous adsorption and reduction) or two steps ex-situ (Figure 17a). The formation of an external iron oxide layer that reduces the risk of further oxidation has been observed on both occasions (Figure 17b). The extra dendritic lamellae ameliorated immobilization and stabilization of the ZVI particles that supported malachite green mineralization for at least eight consecutive cycles (Figure 17c) [152]. Alternative activation procedure of the PET fibers, alkaline hydrolysis, and optional PAMAM binder layer permitted uniform reduced GO coating and high Fe^0^ loading after dip coating in Fe^3+^ and GO solutions and in-situ reduction-immobilization (Figure 17d). The produced non-woven fabrics were used for the removal of crystal violet (Figure 17e) [153]. In contrast, PAMAM G1 proved a rather incompatible host for Fe NPs since its attachment to polyester fabrics yielded inferior 4-nitrophenol and methylene blue degradation rates in comparison to APTES or thioglycerol [154].

The majority of the heterogeneous catalysis systems though are in the form of microspheres or beads. The superiority of dendritic polymers-metal NPs over linear analogs was established by a direct comparison of poly(styrene)-co-poly(vinyl benzene chloride) cores surrounded by three different types of polymer-shells (i) triethanolamine, (ii) glycidyl trimethyl ammonium chloride, and hyper-branched polyglycerol. Regardless of the metal NPs (Ag or Au) Congo Red was processed (Figure 18a) at least twice faster in the suspension of the catalyst with the hyperbranched polymer exterior in comparison to the linear nitrogen-containing analogs [155]. Au NPs stabilized with PPI G2, G3 based on Poly(styrene)-co-poly(4-vinyl pyridine) divinylbenzene beads exhibited the best performance over Ag, and Pd for reduction of trypan blue (Figure 18b) [156]. PPI G2, G3 were also grafted on poly(vinyl imidazole) microbeads prepared by suspension co-polymerization of styrene, N-vinylimidazole, and divinylbenzene. When bimetallic Au/Pd NPs were immobilized into the dendrimers they exhibited 3-fold superiority in comparison to monometallic Au against malachite green and unaltered performance over five catalytic circles after recovery by simple filtration [157].

Apart from metal NPs matrices, dendritic polymers may also act as templates for the supporting nanospheres. PAMAM G4 dendrimers bearing Au and Ag NPs molded within their cavities acted as a dual templating agent by further constructing external mesoporous silica nanosphere shells. After removing the organic content by calcination at 500 °C and a preliminary survey with p-nitrophenol oxidative degradation of methylene blue was observed with very little leaching. Preparative oxidation of benzyl alcohol was also attained [158]. A similar configuration with Ag NPs and hyperbranched PEI in the place of PAMAM established the capability of further water disinfection from *Staphylococcus aureus*, *Pseudomonas aeruginosa* and *Escherichia coli* bacterial load as well [159]. Onisuru presented a second way to take advantage of this dual templating mechanism. At first PAMAM G4 enfolded Cu NPs then a mesoporous SiO_2_ integument was created. The organic scaffold was pyrolyzed by calcination at 500 °C. Displacement of Cu by galvanic exchange with Au led to nanospheres able for 4-nitrophenol reduction, and oxidation of rhodamine B and styrene (Figure 19) [160].

The field of dye elimination and subsequent recovery of magnetic beads took also an advantage of the initial evolution based on experiments on nitro-derivatives. The productive merging of divinylbenzene and glycidyl methacrylate coating was upgraded with the inclusion of 4-methyl styrene. Suspension polymerization led to a copolymer, coating for 8 nm magnetite cores. Grafting PAMAM dendrons G2 doped with Au NPs proved 6 times more effective for rhodamine B discoloration than employing G0 analogies for five consecutive cycles. [161]. The heterogeneous catalysts suitable for pigment discoloration are listed in Table 5.

### 3.3. Applications of Heterogenous Catalysis in the Treatment of Other Water Contaminants

Recently, efforts to address water contamination issues caused by toxic compounds that do not come under the dyes and nitroaromatic derivatives categories have begun. All these efforts concern equipping filtration membranes with the ability to decompose the absorbed pollutants and interestingly all involve bimetallic NPs. A PPI β-cyclodextrin derivative was synthesized by a typical conjugation reaction from β-cyclodextrin carbonyl imidazole. Interfacial polymerization by dip coating of commercial polysulfone microporous membranes into an aqueous solution of this dendrimer and subsequent crosslinking with trimesoyl chloride afforded reactive films. These membranes adsorbed Fe^2+^ and Ni^2+^ that amalgamated into bimetallic Fe/Ni NPs with the aid of NaBH_4_. The final membranes managed dichlorination of adsorbed 2,4,6-trichlorophenol to 2,4-dichlorophenol,4-chlorophenol, and phenol [162]. Polyethersulfone support membrane modified by simple PPI G2, G3 hosted Fe/Ni NPs induced by the same process. When standard solutions of three absorbed arginine-containing microcystin congeners (MCY-LR, MCY-YR, and MCY-RR) were filtered by the combined membrane, degradation greater than 80% was observed. [163] Replacement of PPI with its non-symmetric PEI analog on polysulfone and selecting a different bimetallic pair Fe/Pd of 4/1 ratio (Figure 20a) carried out the dechlorination of adsorbed polychlorinated biphenyl-153 (Figure 20b). The minimal metal leaching was observed after four days was attributed to the solid incorporation of the NPs into the internal hyperbranched cavities [164].

## 4. Unconventional Catalysis Involving Dendritic Polymers

Although the vast majority of both conventional homogenous and heterogenous catalytic pollutant degradation concerning dendritic polymers is accomplished with the inclusion or the stabilization of metal NPs the are some fascinating exceptions. The minimal deviation from the norm is the omission of the cation reduction step. In a first example, common polyacrylonitrile fibers were molded with polyamines (ethylenediamine, tetraethylenepentamine, and hyperbranched PEI) susceptible to complex formation with Fe^3+^ and were applied for synthetic reactive dye wastewater (including reactive red 195) discoloration [165]. Chitosan is contemplated as a second intriguing option since it is a biocompatible renewable organic carrier with a very advantageous molecular architecture. In the form of dendronized microbeads covalently bonded with hexa- and nona-functionalized aromatic or aliphatic polyamido-dendrons (Figure 21a), it can retain Cu^2+^ ions and form a multipurpose catalyst capable of promoting decomposition of H_2_O_2_ for water and soil organic pollutant oxidative degradation [166].

Besides metal NPs and ions, PAMAM G3 cavities may envelop other inorganic materials such as montmorillonite or rice-straw-ash NPs (RSA) and magnetite NPs (Figure 21b). Increasing the percentage of the latter from 0.4 to 1.2 wt.% causes clustering of the organic scaffolds to “dendrimer colonies”. As a secondary side effect coalescence of SiO_2_ NPs of RSA from ~53 to 180–650 nm and exfoliation of clay layers is observed. These two hybrids although designed for optimal adsorption properties (NH_4_^+^, NO_3_^−^ for magnetite @PAMAM/montmorillonite), (Hg^2−^, Br^−^ for magnetite @PAMAM/rice straw ash) also displayed oxidative degradation potential to xylenol-orange (~85%) and malachite-green (~99%) respectively. Mineralization in the first instance was justified by the interactions of the dye with the chemical linkage of magnetite NPs to amidic groups of PAMAM whereas in the second case by the anchorage of magnetite NPs to the terminal amine groups [167].

Templating properties of dendritic polymers for inorganic ceramic formations were exploited for the manufacturing of SiO_2_@TiO_2_ core-shell nanospheres. Conventional hydrolysis-condensation of tetra ethoxy silane created the spherical core. Following calcination at 500 °C, hydrolysis of titanium isopropoxide created a homogenous titania coating. Apart from the photooxidation of NO the nanospheres carried out 4-nitrophenol reduction [168]. Replacing the external TiO_2_ overlay by a shell of CeO_2_ cubic nanocrystals with sizes ranging between 2 and 6 nm produced a similar result [169]. A further example where templating by a dendritic polymer plays a substantial role in catalytic properties is the synthesis of CuFe_2_O_4_ nanoscale-confined precursor via coprecipitation of Fe^+3^ and Mn^3+^ nitrates with a hyperbranched polyamide. Restriction into the constrained dendritic cavities generates much smaller particles with higher specific surface area in comparison to the ordinary CuFe_2_O_4_. This effect is reflected in the Fenton oxidation output of several dyes (reactive red 2, reactive yellow 3, basic red 46, and basic yellow 24) and is so profound that this technique proved superior to the calcination treatment [170]. Hyperbranched phenylene diamine methyl methacrylate HBPDMMA G2 exerts equivalent templating function to the coprecipitation of ferrous and manganese cations in pH = 9 directing the formation of layered double hydroxides (G2/FeMn-LDH) (Figure 22a). Subsequent calcination yields ferromanganese oxide (FeMnO_3_) suitable activator to peroxymonosulfate for methylene blue, tetracycline, and rhodamine b decomposition [171].

Closing this section, two commendable efforts for the formation of complexes between dendritic polymers and enzymes for water purification purposes are worth mentioning. The first concerns enveloping laccase enzyme from *Trametes versicolor* with an amphiphilic dendritic linear dendritic ABA dendronized copolymer consisting of linear polyethylene glycol and poly (benzyl ether) dendrons. This treatment enhanced the biodegradation capabilities of the enzyme towards bisphenol. The effect was more pronounced with G2, G3 dendron generations while stereochemical restrictions were observed with G4 [172]. Hyper-branched PAMAM was further combined with hyaluronidase (Figure 22b). The attachment was performed by adsorption in phosphate buffer solution. The resulting nano-bio catalyst proved ideal for the enzymatic degradation of hyaluronic acid exceeding the performance of the bare enzyme [173]. All these unorthodox examples are summarized in Table 6. 

**Figure 22 nanomaterials-12-00445-f022:**
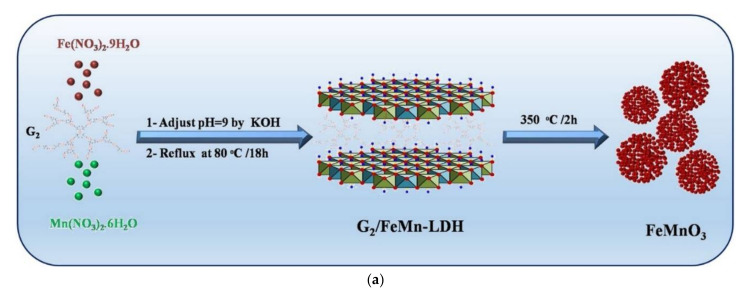
(**a**) Synthesis of FeMnO_3_ through the calcination of its precursor G2/FeMn-LDH. Reproduced with permission from [171]. (**b**) The 3D position of the minimum docking energy pose of PAMAM with Hyaluronidase enzyme, and representation of interactions (hydrogen bonds and electrostatic binding) of PAMAM (blue color, stick) in the enzyme. Reproduced with permission from [173].

## 5. Photocatalytic Decomposition

### 5.1. Photocatalysis on the Prominent TiO_2_ Substrate

Since the discovery in 1972 by Fujishima and Honda that water splits photocatalytically when in contact with TiO_2_ electrodes [174], titania is recognized as one of the most auspicious photocatalysts [175]. As expected, research on the dendritic polymers’ potential involvement in aqueous pollutants photocatalytic degradation was initially based on TiO_2_ and was conducted almost in parallel with conventional catalysts. NPs. PAMAM G4, PAMAM-OH G4, PAMAM-COONa G4,5 acted as templated agents for the development of TiO_2_ NPs confined into the dendrimer cavities. This was achieved by hydrolysis of TiCl_4_. Smaller sizes were observed (4.4–6.7 nm) in comparison to the method without dendrimer mediation (7.5 nm) and this was reflected in their better efficiency on the photodegradation of 2,4-dichlorophenoxyacetic acid [176]. Even smaller (1–5 nm) TiO_2_ NPs were synthesized at ~0 °C by hydrolyzing [(CH_3_)_2_CHO]_4_Ti in 1-propanol. PAMAM dendrons with a triethoxysilyl focal point and hexyl chains at their periphery (Figure 23a) successfully inhibited aggregation and reinforced photocatalytic decomposition of 2,4-dichlorophenoxyacetic as well. As confirmed by X-ray photoelectron spectroscopy this was realized by the formation of Si–O–Ti covalent bonds [177]. Aromatic polyamide dendrimer pairing with TiO_2_ NPs also yields an improved hybrid photocatalyst for phenol under visible light irradiation in comparison to bare titania. Functionalization by spiro lactam molecular switch further tendered pH-responsive capability. At pHs below 3 spiro lactam transforms to the ring-opened amide (Figure 23b) form which absorbs more intensely and exhibits even better activity [178]. TiO_2_ nanowires (NWs) were shaped too by a hyperbranched polyester with hydroxyl terminal groups. The reaction took place in an alkaline medium by hydrothermal treatment of TiO_2_. NPs Photocatalytic mineralization of wastewater proceeded much faster with these very active hybrids than the respective simple NWs [179].

As is the case with conventional catalysts magnetic photocatalysts may equally derive from dendritic polymers with Fe_3_O_4_ core. Instead of a templated core nano-sized titania terminals were linked to the periphery of PAMAM dendrons propagated by the divergent method. Alongside the advantage of convenient magnetic separation, discoloration of methyl orange proceeded more promptly in comparison to bare nano TiO_2_ or Fe/Ti mixture. Dendrons cavities act as adsorbents that are “regenerated” by the catalytic decomposition of the pigment on the TiO_2_ NPs [180]. Methyl orange mineralization effectiveness grows with increasing generation number due to the larger number of external nano-TiO_2_ NPs and their smaller size. On the other hand, in multiple reaction cycles, PAMAM G3 composites were losing their activity more rapidly [181]. The combination of TiO_2_ with CdS NPs (Figure 23c) produced an even better photocatalyst than TiO_2_ NPs alone as established by calculated apparent quantum yields (3.6 × 10^−5^ molecules photon^−1^) and figures of merit (100) [182].

Besides the role of matrices, there are simpler prospects for fruitful interaction between dendritic polymers and TiO_2_. Polycrystalline calcinated TiO_2_ was impregnated with a dendrimer with zinc phthalocyanine core bearing poly (arylbenzyl ether) branches. This process improved photolytic capacity for rhodamine B to 97% (compared to both titania about 65% after 5 h and zinc (II) phthalocyanine complex about 90% after 5 h [183]. In a second example, PAMAM G4 was used to exfoliate nanoclay modified by an amphiphilic quaternary ammonium salt; nano TiO_2_ was added to introduce photocatalysis properties to the absorbent. Polyester fabrics were immersed to this dispersion and impregnated by the pad-dry-cure method (Figure 23d). The composite textiles successfully discolored Reactive red 4 solutions without getting colored themselves [184]. A very simple, yet extremely interesting idea is doping titania by Pt and N atoms by a modified sol-gel method using PAMAM G0, G1, and PEI for platinum nucleation and as sources of nitrogen. After calcination, the resultant ceramic photocatalyst was able to eliminate brilliant black (99.83%) in 3 h by irradiation with visible light. Higher activity enhancement and a more profound reduction of the bandgap were observed with the PAMAM G1 matrix [185]. 

**Figure 23 nanomaterials-12-00445-f023:**
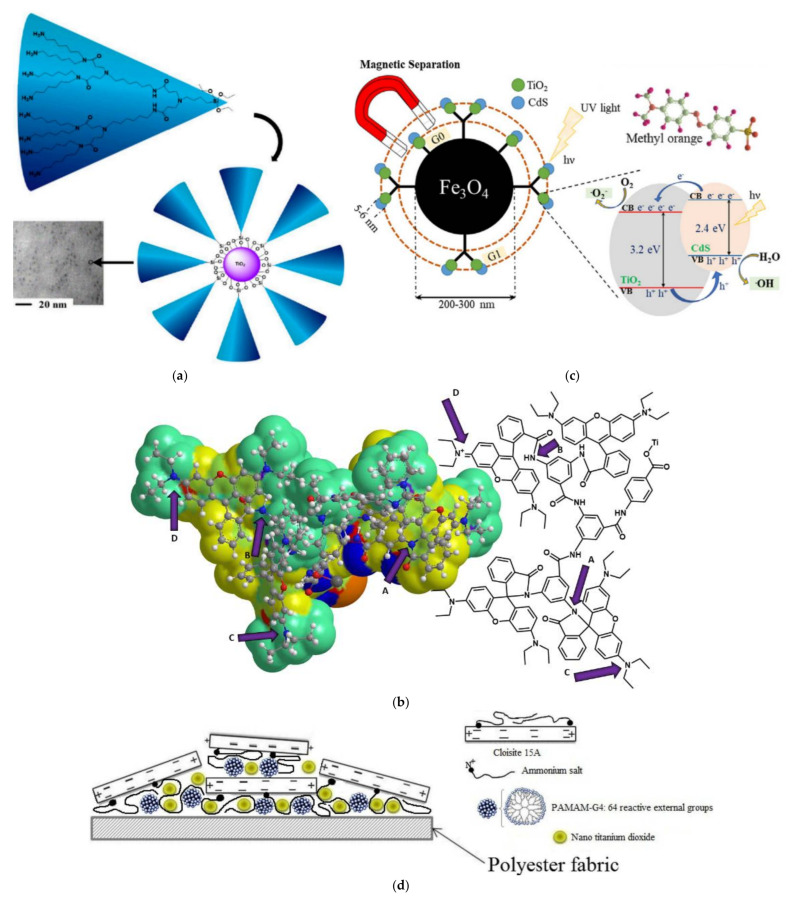
(**a**) Pathway for the clipping of the PAMAM dendron protective layer to titania photocatalytic nanoparticles and SEM micrograph of the complexes. Adapted with permission from [177]. (**b**) Chemical formula and partial charge molecular model (yellow to red positive, cyan to blue negative) of two dendritic branches depicting the two modes of the molecular switch: spirolactam, ring-closed (A) amine external groups (C) (switch off) amide, ring-opened (B) aminium external groups (D) (switch on) Adapted with permission from [178]. (**c**) Schematic representation of TiO_2_/CdS nanocomposite stabilized on a Fe_3_O_4_ core by PAMAM G0, G1 dendrons, and the photolytic mechanism for methyl orange. Reproduced with permission from [182] (**d**) Schematic of polyester fabric coated with three components of nano clay, nano TiO_2_, and PAMAM dendrimers. Reproduced with permission from [184].

### 5.2. Photocatalysis with Other Photoactive NPs

MWCNT technology initially developed for 4-nitrophenol treatment was adapted for photocatalysis too. Nanosized (3–5 nm) spherical semiconductor CdS particles were fastened onto aryl modified MWCNTs by PAMAM-COOH dendrons (Figure 24a). Under visible light irradiation, MWCNTs accumulate photo-induced electrons, transferred from CdS. Ensuing reaction with oxygen generates O^2•−^ superoxide radicals that are transformed to very reactive HO^•^ and HOO^•^. In synergy with the holes created in CdS, they substantially exceed the performance of simple reference CdS NPs towards rhodamine B degradation [186]. In the allotropic layered reduced graphene oxide (RGO) hyperbranched polyamide-amine fulfilled the binary role of binder to dialdehyde cellulose fibers and of templating nanoreactor for the nucleation of semiconductor copper sulfide NPs (Figure 24b). The outcome was a composite paper able for efficient photolytic elimination of rhodamine B about 2.7 times faster than simple dialdehyde cellulose with CuS. This was associated with improved charge separation efficiency since RGO effectively suppresses the recombination of photogenerated carriers. The solid attachment of RGO and the uniformity of CuS were also credited for this higher activity that was retained up to 90% after 10 cycles [187]. The reverse concept is also feasible and practical. Electrospun nanofiber mats obtained from pullulan/poly (vinyl alcohol)/poly(acrylic acid) were impregnated into ceria suspensions. PAMAM G3 was attached to the immobilized CeO_2_ NPs with the aid of glycidyl triethoxysilane (Figure 24c). This addition shifted the bandgap of the semiconductor towards visible light for the photolysis of phenol, 4-Hydroxy-1-naphthalene sulfonic acid sodium salt, and azorubine dye [188].

Self-Assembly of anionic polyoxometalate (POM) K_4_[SiW_12_O_40_] -PAMAM G4 agglomerates by electrostatic interactions in 100 nm coil-like structures (Figure 24d) produced a promising photocatalyst exhibiting electrostatic selectivity. Best results were obtained by cationic dyes methyl red and methylene blue in comparison to neutral alizarin yellow R and in contrast to anionic xylenol orange which is protected from POM photolysis by the dendrimer [189]. Simple templating of MoS_2_ by reduction of sodium molybdate by thiourea in the presence of PAMAM G2 yielded monodispersed and well-distributed NPs that reduced more than 95% the percentage of two commonly used pesticides, chlorpyrifos and glyphosate in water (Figure 25) [190]. Hydrothermal preparation of bismuth vanadate into the cavities of hyperbranched PEI on top of affecting NPs size also changes the monoclinic lattice organization of BiVO_4_ to 72% tetragonal, 28% monoclinic, reduces the electron-hole recombination rate and the bandgap. Preparation of modified polyethersulphone membranes incorporating these PEI-BiVO_4_ complexes via the phase inversion method improved water flux and hydrophilicity and allowed the degradation of triclosan up to 86%. [191]. 

Adsorption of PAMAM-COOH G2.5 by ZnO/CuS low-dimensional heterostructured functional materials afforded a hybrid displaying potent photoresponse properties that were further enhanced by doping with GO and verified by processing rhodamine B and methylene blue [192]. The same ex-situ mixing technique was adopted for the surface modification of ZnO nanorods by hyperbranched polyester and was accompanied by a second in-situ method where the polycondensation reaction for the synthesis of the polymer took place in the presence of ZnO. Both methods increased photocatalytic degradation of phenolic organic pollutants (N-(1-naphthyl)-ethylenediamine dichloride, catechol, and sulfanilamide) and antibacterial activity against Gram-positive *Bacillus cereus*, and Gram-negative *Escherichia coli*. The conventional adsorption method exhibited the best results [193].

On the other hand, the chemical binding of dendritic polymers to ZnO is accomplished in the same way as SiO_2_ and Fe_3_O_4_ oxides. Glycidoxypropyltrimethoxysilane conjoined PAMAM G4 to Wurtzite nanorods and rendered them more efficient to azo dye naphthol blue-black disintegration under UV light than bare ZnO nanorods [194]. In another rather more sophisticated way, starch-modified hyperbranched polyol was copolymerized by an in-situ polymerization process with a prepolymer composed from polycaprolactone diol, 2,4/2,6-toluene diisocyanate, and 1,4 butanediol to give the respective polyurethane (Figure 26a). This resulting dendrigraft was further crosslinked by varying amounts of zinc oxide merged with reduced carbon dots (RCD). The optical bandgap of the final composites (2.12 eV) was significantly lower to RCD (3.18 eV) and ZnO (3.3 eV) and this decrease was mirrored to their photocatalytic potential towards aqueous surfactant contaminants, dodecyl benzenesulfonate (96.7% in 110 min) and commercial detergent (94.8% in 150 min) [195]. At this point, it is interesting to note that photocatalytic substrates such as ZnO frequently present synergistic pollutant absorption properties [196]. The selection for the presentation of the last example of this review is in our opinion the most impressive one. It is about a dendritic polymer hyperbranched polyimide with inherent photocatalytic properties that are magnified by the oxidative introduction of N-O bonds to the triazine core (Figure 26b). The transformation to polyimide N-oxide changes dramatically the electronic structure of the conjugated polymer. The localization of the highest occupied molecular orbital is shifted from the melamine groups to the N-O bonds (Figure 26c) which are also the main adsorption sites. This synergistic adsorption-photocatalytic decomposition mechanism proved very drastic towards the elimination of methyl orange [197]. All photocatalysts comprising dendritic polymers that were employed for aqueous pollutant elimination are summarized in Table 7. 

## 6. Conclusions

It is evident that despite the multitude of efforts and examples presented herein, the research field of pollutant disintegration by the involvement of dendritic polymers is still in its early adolescence compared to the maturity of the adsorption alternatives since the majority of the systems proposed have only been tested in simple catalytic reactions. In addition, continuous mode treatment through columns is implemented in just one case; and scaling up and application of the purification layouts in actual wastewater management in static batch reactors or fixed beds has not been tried so far. Yet pairing dendritic molecules with catalytic and photocatalytic ingredients proved very beneficial as it enhanced their activity, stability, and overall properties. They offer a substantial advantage since smaller quantities can address far heavier water pollution incidents due to easier regeneration in contrast to the adsorbents. They may also act synergistically with adsorbents inhibit their poisoning and prolong their use. Collation with other NPs templating agents reveals both their metal nucleation and pollutant decomposition potential. As general rules of thumb, the size of the NPs can be manipulated by specific functional groups in the internal dendritic cavities and the host-guest stoichiometry. Particle size and specific surface area play more important roles than crystallinity. Smaller is better leading to increased activity and faster reaction rates. Bulky dendritic layers repeatedly caused reactant diffusion delays.

Future perspectives in the field of basic research could be based on the following axes (a) Thorough research of the formation mechanisms and catalytic properties of metal NPs. Some metals such as Ni, Rh, Co, Ir, with established catalytic and antimicrobial potential have not received deserved interest [198,199] (b) Expansion of the investigation of more semiconductors that are capable of photolyzing water contaminants (c) Application of bimetallic or polymetallic nanoparticles and dual dendritic polymer functionalities (d) Implementation of multifunctional composites capable of simultaneous adsorption, catalytic degradation, and disinfection. (e) Expansion of the test to other classes of pollutants. Pesticides, surfactants, and antibiotics for instance have not attracted enough research.

## Figures and Tables

**Figure 1 nanomaterials-12-00445-f001:**
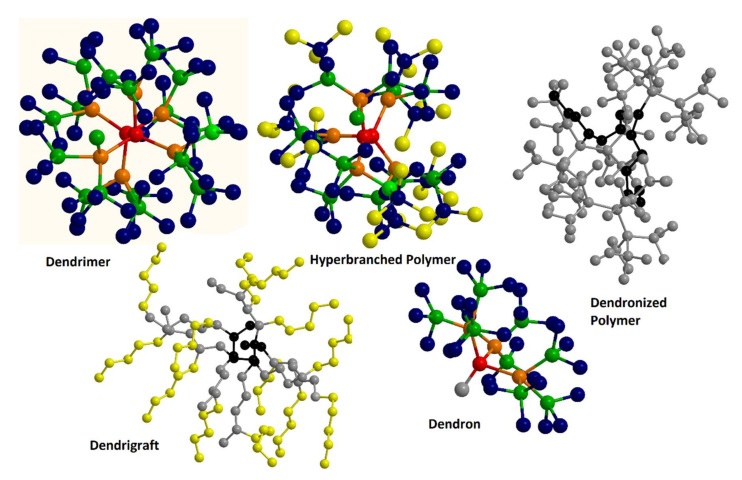
Schematic representation of the categories of dendritic polymers. Each sphere represents a monomer. Different colors depict the different polymerization generations.

**Figure 2 nanomaterials-12-00445-f002:**
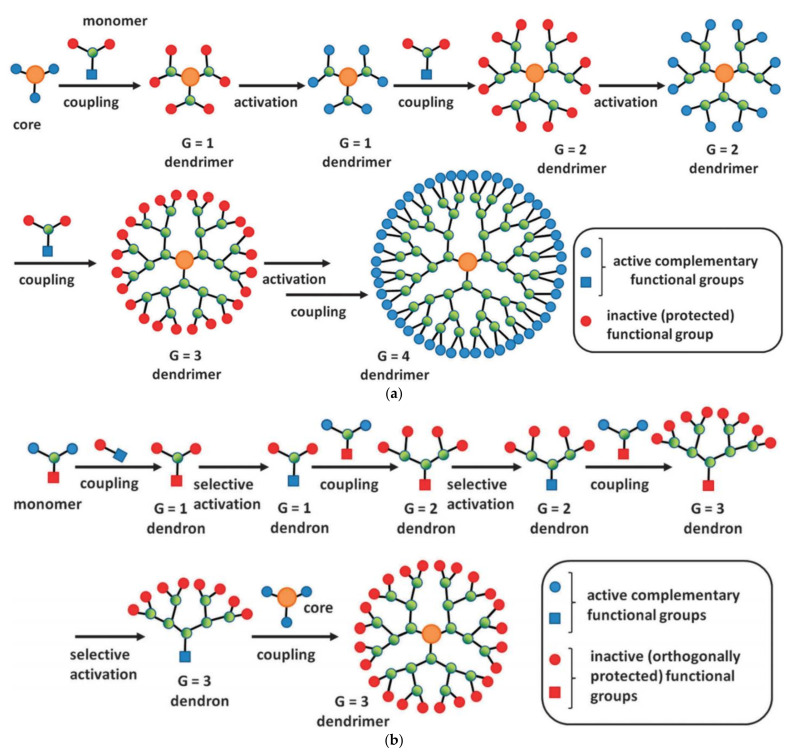
(**a**) Synthesis of dendrimers according to the divergent method. (**b**) Synthesis of dendrimers according to the convergent method (reproduced with permission from [39]; Copyright, Royal Society of Chemistry).

**Figure 3 nanomaterials-12-00445-f003:**
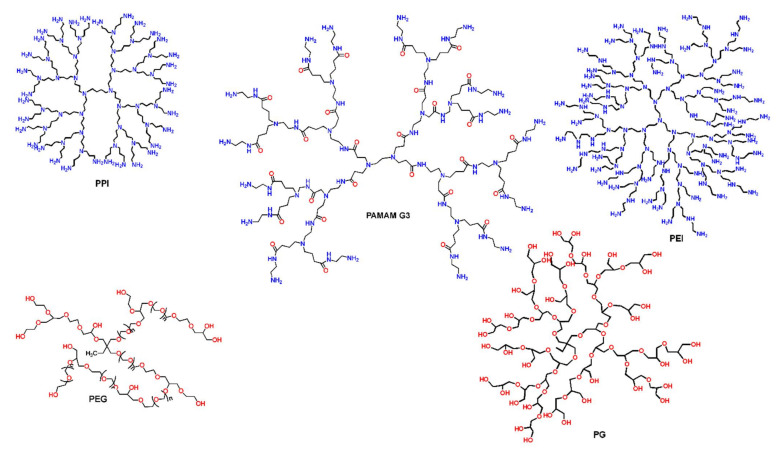
Chemical formulas of the most common dendritic polymers.

**Figure 6 nanomaterials-12-00445-f006:**
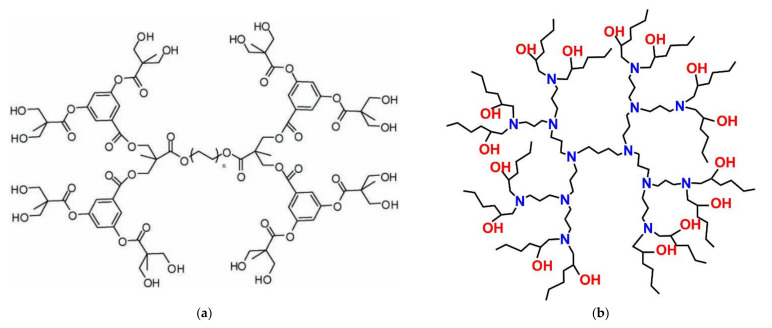
(**a**) Chemical Structures of PEG-G1-(3,5-DHB-OH)16 dendrimer. Reproduced with permission from [93]. (**b**) Structure of amphiphilic PPI G2 dendrimer.

**Figure 7 nanomaterials-12-00445-f007:**
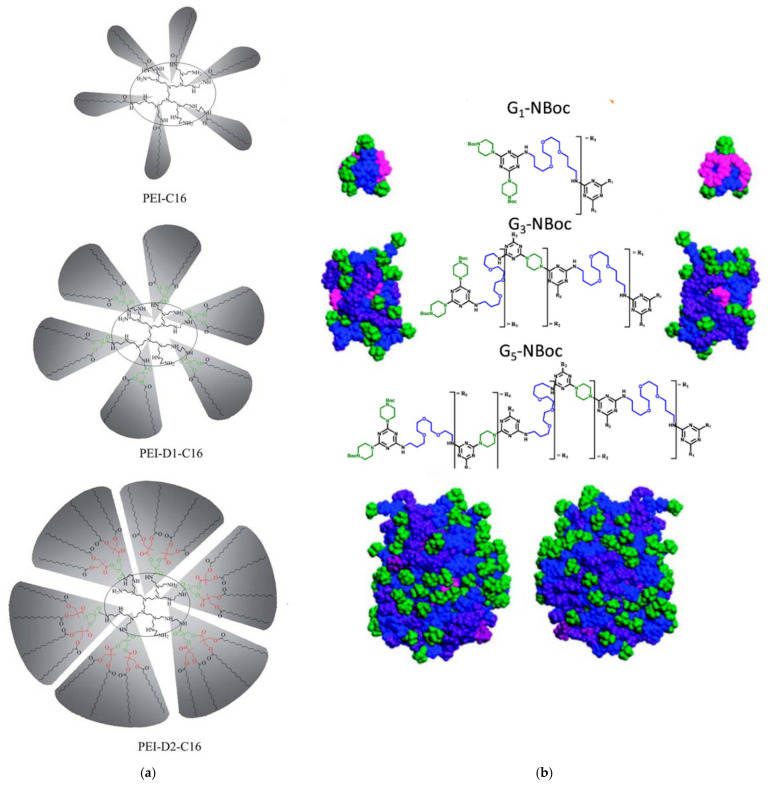
(**a**) Amphiphilic hyperbranched PEI copolymers with different shell morphologies bearing 1, 2, or 4 palmitamides per external amino group of the parent dendritic core: Au NP templates and stabilizers Reproduced with permission from [96]. (**b**) Chemical structures of G1, G3, G5 triazine-based dendrimers and front and back sides of dendrimers G1-NBoc, G3-NBoc, and G5-NBoc with Boc groups (green), peripheral triazines (blue), outer generation (purple), inner generation (raspberry), and core (salmon) indicated by color. Reproduced with permission from [102].

**Figure 8 nanomaterials-12-00445-f008:**
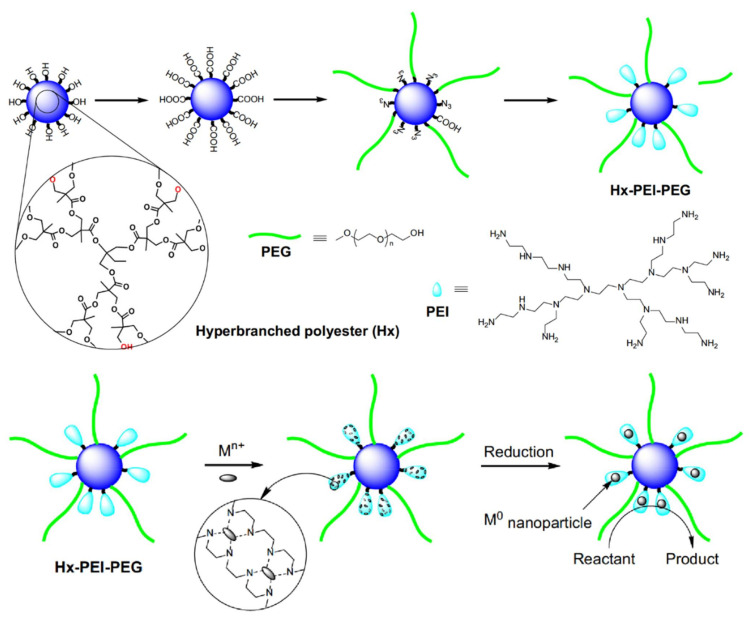
Schematic representation of the synthetic route for the assembly of the composite polymers Hx-PEI-PEG and the subsequent reduction of Au ions into PEI cavities. Reproduced with permission from [103].

**Figure 9 nanomaterials-12-00445-f009:**
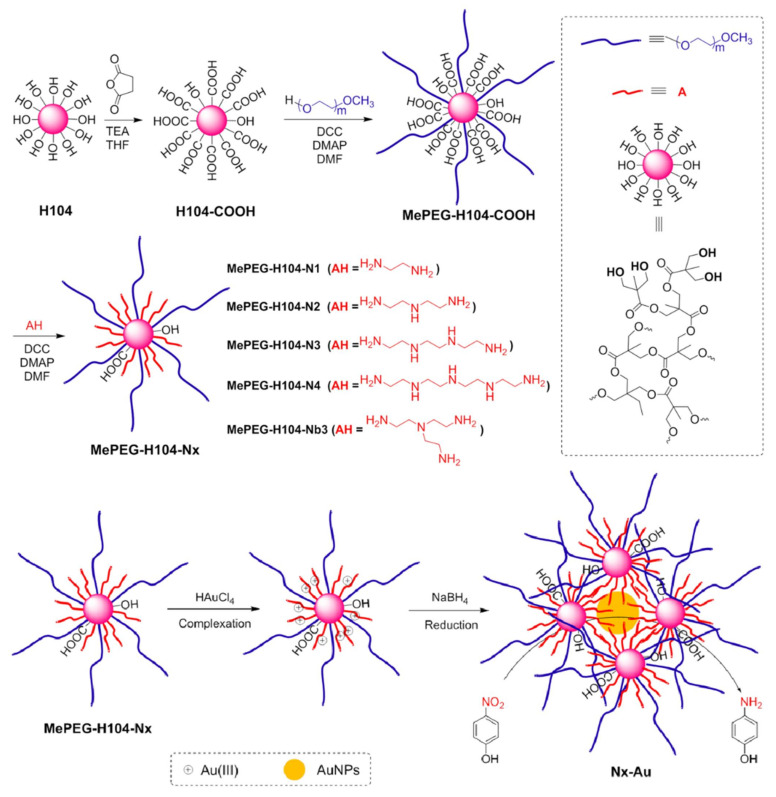
Synthesis of the amine-modified copolymers H104-PEG-Nx and the subsequent reduction and stabilization of Au ions. Reproduced with permission from [104].

**Figure 11 nanomaterials-12-00445-f011:**
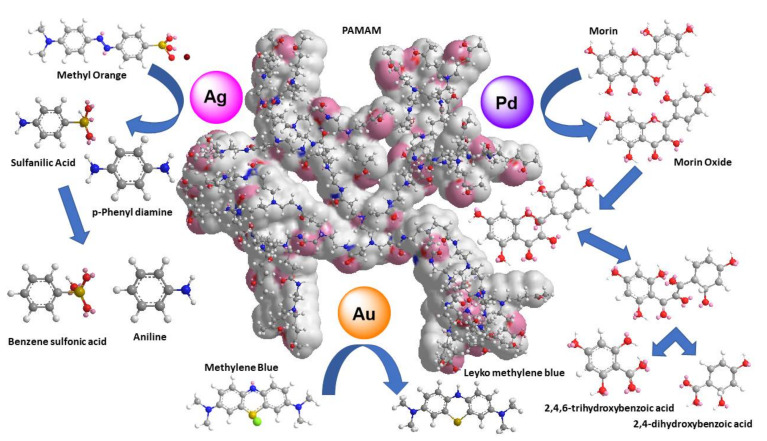
Schematic representation of catalytic decomposition paths of Morin, Methyl Orange, Methylene Blue dyes mediated by metal NPS.

**Figure 15 nanomaterials-12-00445-f015:**
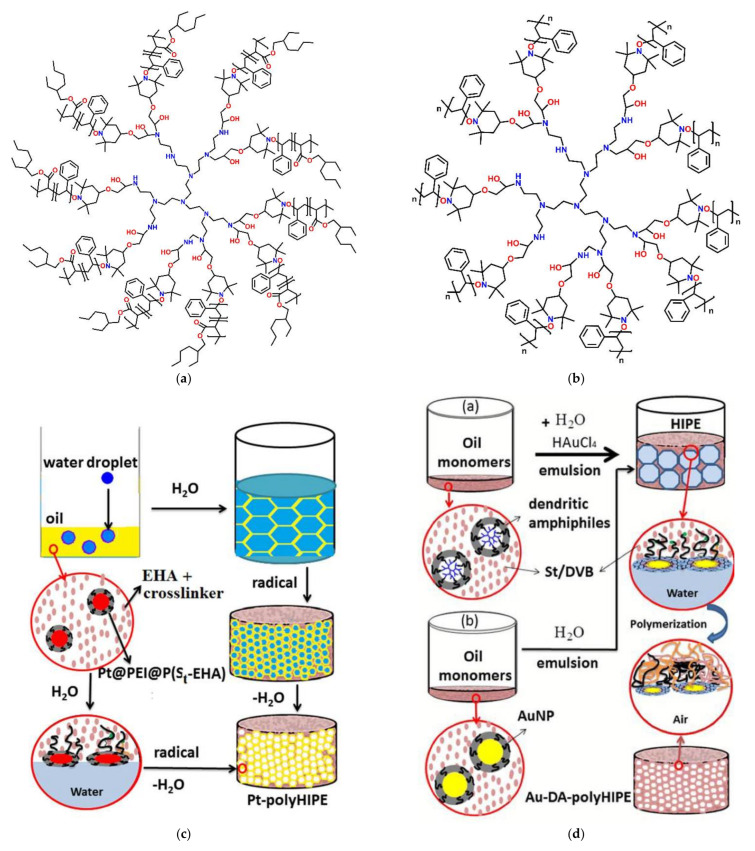
(**a**) PEI Core Poly (styrene-co-2-ethyl hexyl acrylate shell dendritic amphiphile. The synthetic procedure of the Graphene oxide-MnFe_2_O_4_-PAMAM G3-Pd nanocomposites. Adapted from [142]; (**b**) PEI Core Polystyrene do-decyl shell dendritic amphiphile. Adapted from [144]; (**c**) Preparation of Pt-polyHIPE open-cellular and elastic monolith Reproduced with permission from [142]; (**d**) Schematic presentation of the synthesis of Au-DA-polyHIPE open-cellular and elastic monolith. Reproduced with permission from [144].

**Figure 16 nanomaterials-12-00445-f016:**
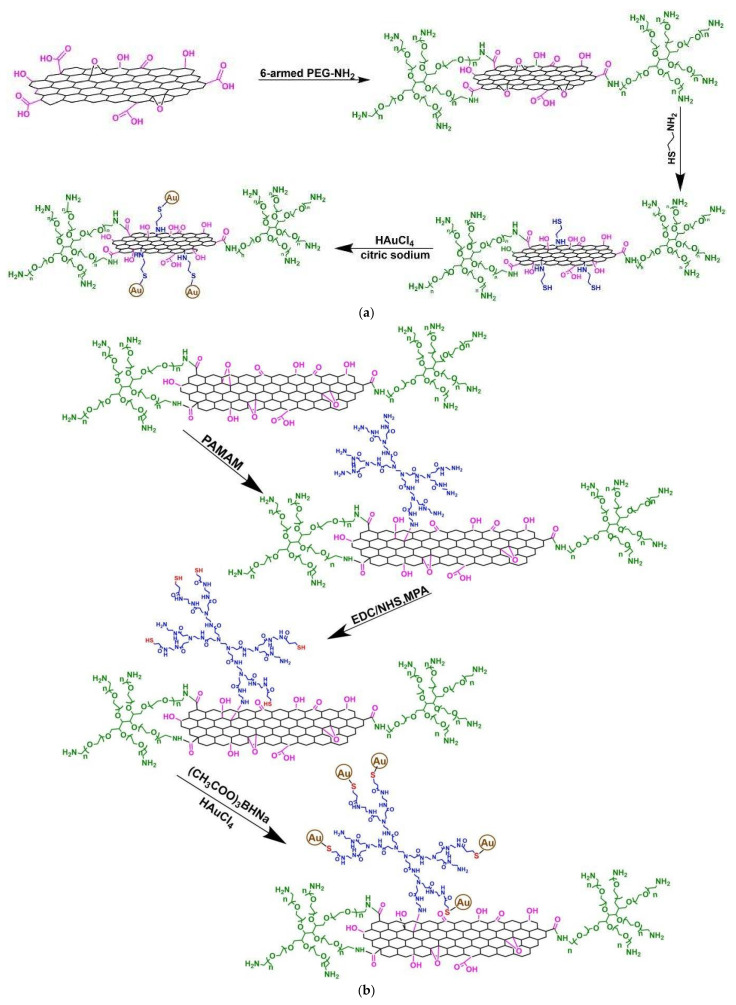
(**a**) Au NPs loading on PEG and thiol dual-functionalized nanographene oxide (**b**) Au NPs loading on PEG and thiol-PAMAM dual functionalized nanographene oxide Both reproduced with permission from [146].

**Figure 17 nanomaterials-12-00445-f017:**
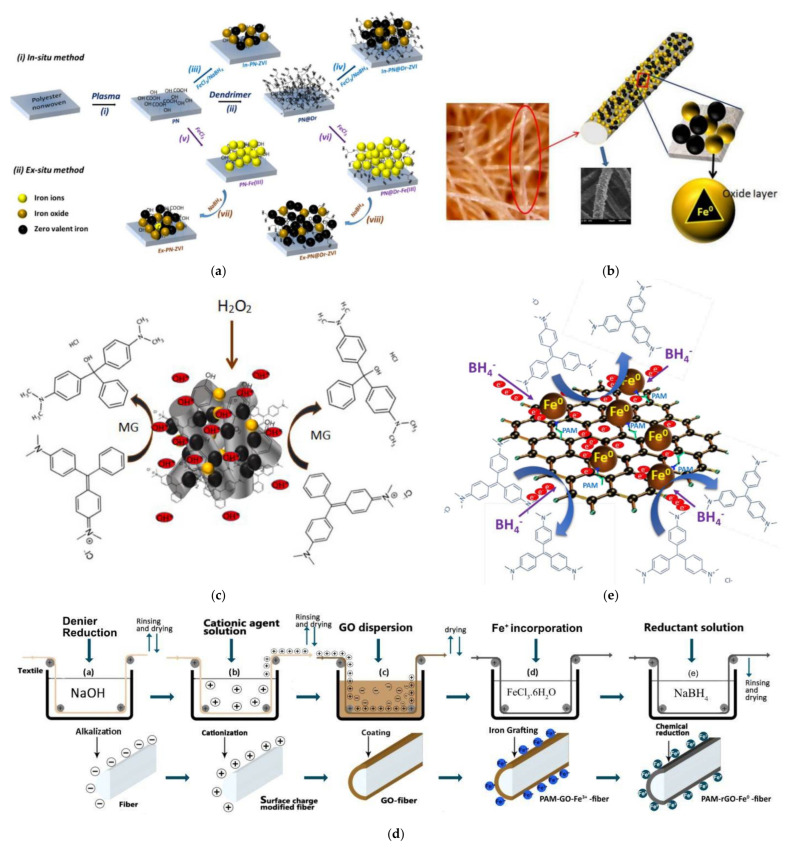
(**a**) Pathways used to immobilize Fe on nonwoven PET, (i) plasma treatment (ii) dendrimer grafting (iii,iv) in-situ and (v/vii,vi/viii) ex-situ reduction method. [Yellow spheres represent iron (III) ions, orange spheres represent ZVI with iron oxides shell, and black spheres represent ZVI]; (**b**) Schematic illustration of Fe^0^ immobilization on PET fibers. (**c**) Mechanism of the catalytic degradation of Malachite Green by ZVI immobilized PET nonwoven. All three reproduced with permission from [152] (**d**) Schematic illustration of (**a**) alkalization of polyester fibers; (**b**) PAMAM-grafting; (**c**) incorporation of GO; (**d**) incorporation of Fe^3+^ (**e**) in-situ immobilization of rGO and Fe0; (**e**) Mechanism for catalytic decomposition of crystal violet dyes using Fe^0^ incorporated polyester nonwoven fabric. Both reproduced with permission from [153].

**Figure 18 nanomaterials-12-00445-f018:**
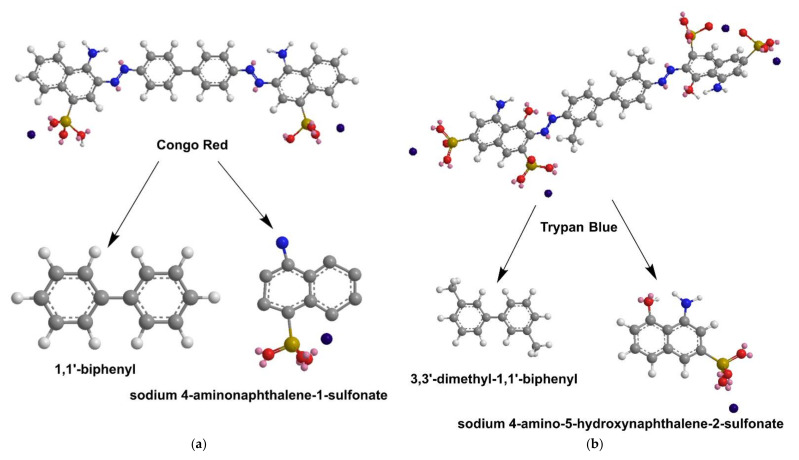
Degradation Paths of Congo Red (**a**) and Trypan Blue (**b**).

**Figure 19 nanomaterials-12-00445-f019:**
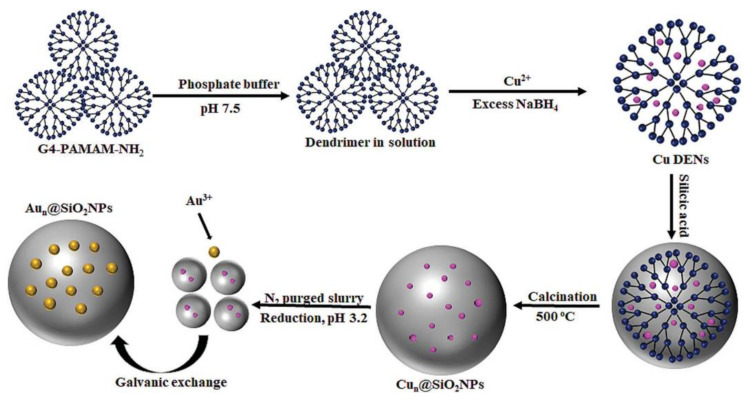
Schematic illustration of Cu NPs immobilization on silica and subsequent in situ replacement by Au NPs. Reproduced with permission from [160].

**Figure 20 nanomaterials-12-00445-f020:**
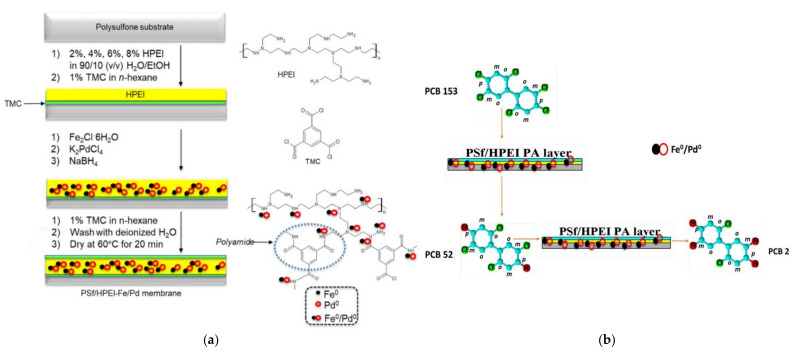
(**a**) Synthesis route for the preparation of Fe/Pd bimetallic nanoparticles embedded in HPEI/PSF membranes. (**b**) Proposed dechlorination reaction scheme. Both reproduced with permission from [164].

**Figure 21 nanomaterials-12-00445-f021:**
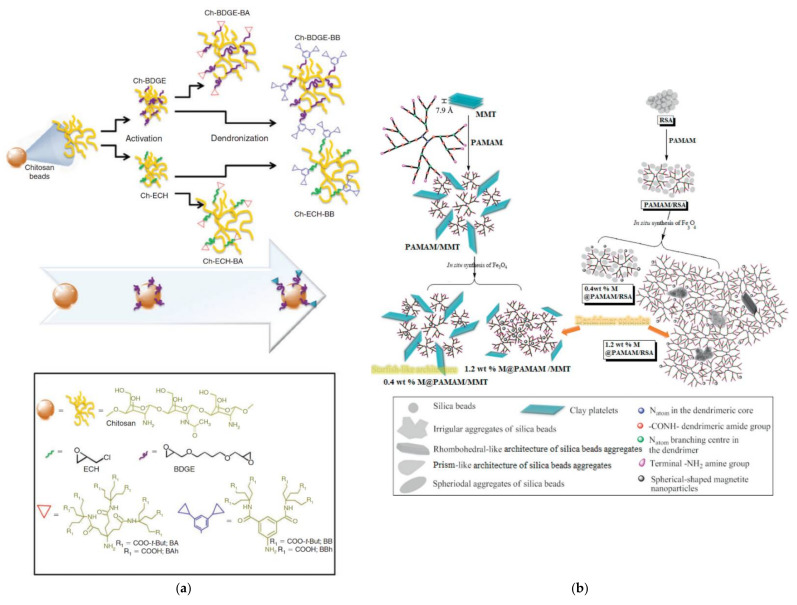
(**a**) Schematic representation of the synthetic routes used for the preparation of the dendronized chitosan microbeads reproduced with permission from [166]. (**b**) Synthetic path for the formation of magnetite @PAMAM/montmorillonite nanocomposites (left) and right magnetite @PAMAM/rice straw ash nanocomposites. Reproduced with permission from [167].

**Figure 24 nanomaterials-12-00445-f024:**
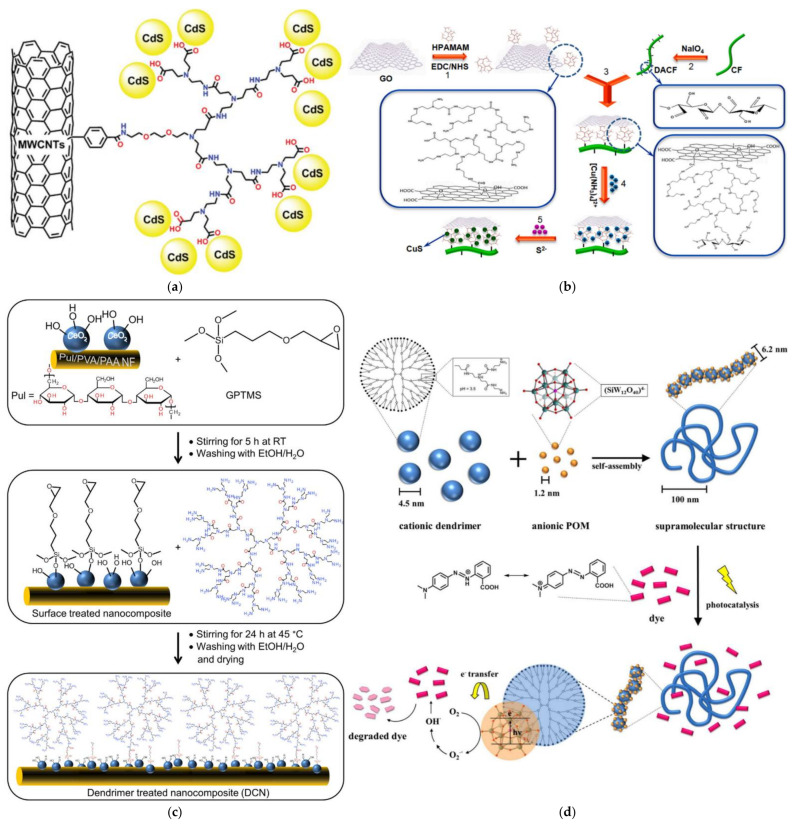
(**a**) Nanosized CdS immobilized onto PAMAM-aryl modified MWCNTs. Reproduced with permission from [186]. (**b**) Synthesis of dialdehyde cellulose fiber/hyperbranched polyamide-amine/reduced graphene oxide/copper sulfide (DACF/HPAMAM/rGO/CuS) hybrid. Reproduced with permission from [187]. (**c**) Surface modification of ceria impregnated nanofiber mats with PAMAM using GPTMS. Reproduced with permission from [188]. (**d**) Schematic representation of POM-dendrimer self-assembly and photocatalysis of methyl red. Reproduced with permission from [189].

**Figure 25 nanomaterials-12-00445-f025:**
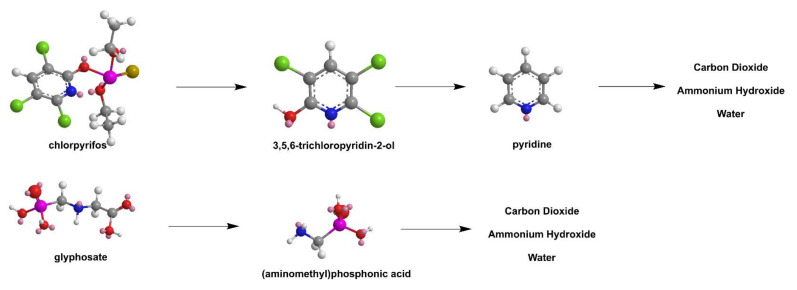
Degradation paths of chlorpyrifos and glyphosate.

**Figure 26 nanomaterials-12-00445-f026:**
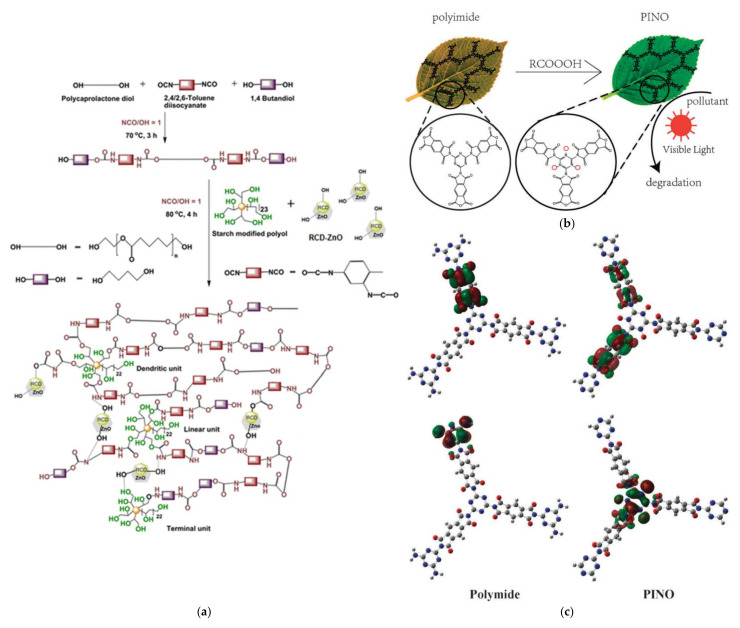
(**a**) Synthetic route for the polyurethane/polyol/reduced carbon dot/ZnO crosslinked dendrigraft composite. Reproduced with permission from [195]. (**b**) The N-site oxidation reaction of the hyperbranched polyimide (**c**) LUMO (above) and HOMO (below) of polyimide and polyimide N-oxide. Both reproduced with permission from [197].

**Table 1 nanomaterials-12-00445-t001:** Dendritic polymer-nanoparticle combinations designed for 4-nitrophenol reduction.

Dendritic Polymer	Metal Nanoparticles	Reference
PAMAM G3, G5 PPI G3, G4	Au	[69]
PAMAM G3, G4, G5, PPI G2, G3, G4	Ag, Pt, Pd	[70]
PAMAM G3, G3.5, G5, G5.5	Ag, Au Ag/Au alloy	[79]
PAMAM-OH G6 PAMAM G4	Au, Cu, Pd, Pt, Au/Cu, Pd/Cu, Pt/Cu	[80]
PAMAM G4	Cu	[72]
PAMAM-OH G4	Ru, Ni, Ru/Ni	[81]
[PEG-G1-(3,5-DHB-OH)16]	Ag	[93]
PEI core 2,2-Bis(hydroxymethyl)propionic acid dendrons	Au	[96]
Hyperbranched Polystyrene	Au	[97]
PAMAM-OH G4, G5, G6	Ru	[81]
PAMAM-OH, PAMAM-NH2 G4, G6	Cu, Ag, Au	[74]
PAMAM G4 PAMAM-OH G4, G6	Pd	[73]
Glycodendrimers	Pd	[84]
Glycodendrimers	Au	[83]
PAMAM-OH G4, G5, G6	Pd	[75]
PAMAM-OH G4, G5	Pd	[71]
Amphiphilic PPI-G2	Ru	[95]
PAMAM G4	Ag, Au	[76]
Nona-PEG-branched Triazole Dendrimers	Au	[85]
Glycodendrimers G1-G4	Au	[88]
PAMAM-OH G4, G5, G6	Pd, Pt	[78]
PEI core amide shells	Au	[98]
Hyperbranched Polyester Boltron-PEI-PEG	Au	[103]
Linear PEG-Hyperbranched PEI poly(ε-caprolactone)	Pt	[105]
Bistriazole-based dendritic amphiphile micelles	Ag, Cu	[89]
PAMAM G5-maleic anhydride-cysteamine	Au	[100]
Triethylene glycol-Arene-Triazole click Dendrimers	Fe, Co, Ni, Cu, Ru, Rh, Pd, Ag. Ir, Pt, Au	[86]
Amine-modified hyperbranched polyester PEG copolymer	Au	[104]
PAMAM-OH G4, G5, G6	Cu	[77]
Jeffamine core PAMAM G4	Ag	[94]
1,2,3-Triazolyl Dendronized Polymers	Au	[90]
Ferrocenyl/Triethylele glycol Dendronized Polymers	Au	[91]
Ferrocenyl/Triethylele glycol Dendronized Diblock Polymers	Au, Ag	[92]
4-Carbomethoxypyrrolidone PAMAM G3-G5	Ag	[101]
Triazine-based dendrimers	Ir, Pt	[99]

**Table 2 nanomaterials-12-00445-t002:** Dendritic polymer-nanoparticle combinations designed for nitroaromatic derivative reduction.

Dendritic Polymer	Metal NPs	Pollutant	Degradation Products	Ref.
Polyaryl ether trisacetic acid ammonium chloride dendrons	Pt	*p*-nitrophenol, *o*-nitroanisole, *o-,m-,p*-nitrotoluene	*p*-nitroaniline, *o*-anisidine and *o*-,*m*-,*p*-aminotoluene	[106]
Polyaryl ether trisacetic acid ammonium chloride dendrons	Au/Pt	*p*-nitrophenol, *o*-nitroanisole, *o*-,*m*-,*p*-nitrotoluene, 3-phenoxybenzaldehyde	*p*-nitroaniline, *o*-anisidine and *o*-,*m*-,*p*-aminotoluene, 3-phenoxyphenyl methanol	[107]
Amphiphilic PPI-G2	Au	Nitrobenzene	Aniline	[108]
PEG core dendrimer	Ag	4-nitrobenzaldehyde, nitrobenzene, 4-nitrotoluene, 4-nitroaniline. 4-nitrocatechol, 2-hydroxy-5- nitrobenzyl bromide, 5-hydroxy-2-nitrobenzaldehyde	4-aminobenzaldehyde, aniline, 4-anisidine, 4-phenylalanine. 4-aminocatechol, 2-hydroxy-5-aminobenzyl bromide, 5-hydroxy-2-aminobenzaldehyde	[109]
PEG-PAMAM G3	Cu	4-(4-nitrophenyl) morpholine, 4-(2-fluoro-4- nitrophenyl) morpholine), 4-(4-nitrophenyl) morpholin-3-one	4-(4-aminophenyl) morpholine, 4-(2-fluoro-4-aminophenyl) morpholine), 4-(4-aminophenyl) morpholin-3-one	[110]
PEG-PAMAM G3	Au	4-(4-nitrophenyl) morpholine, 4-(2-fluoro-4- nitrophenyl) morpholine), 4-(4-nitrophenyl) morpholin-3-one	4-(4-aminophenyl) morpholine, 4-(2-fluoro-4-aminophenyl) morpholine), 4-(4-aminophenyl) morpholin-3-one	[111]
PAMAM G2	Cu	4-nitrophenol, 2-nitrophenol, 4-nitrobenzaldehyde, 2,4 dinitrophenol, 2-nitroaniline, 4-nitroaniline, 3-nitrotoluene, 4-nitrotoluene, 4-nitrochlorobenzene	4-aminophenol, 2-aminophenol, 4-amino benzaldehyde, 2-nitro-4-aminophenol, 2-phenyl diamine, 4-phenyl diamine, 3-aminotoluene, 4-aminotoluene, 4-aminochlorobenzene	[112]

**Table 3 nanomaterials-12-00445-t003:** Dendritic polymer-Nanoparticle combinations designed for the treatment of dyes.

Dendritic Polymer	Metal NPs	Dye	Degradation Products	Reference
Octyl PPI-G2	Ag, Pd, Pt	Methyl Orange	(a) Sulfanilic acid (b) N,N-dimethyl, p-phenylenediamine (c) Benzenesulfonic acid, aniline	[113]
PAMAM-OH G6	Pd, Pt	Morin	(a) Substituted benzofuranone (morin oxide) 2,4-dihydroxybenzoic acid, 2,4,6-trihydroxybenzoic acid	[117]
G6-PAMAM-NH_2_	Au	Morin	(a) Substituted benzofuranone (morin oxide) 2,4-dihydroxybenzoic acid, 2,4,6-trihydroxybenzoic acid	[116]
PAMAM PAMAM-OH G4, G5	Pd, Au	Methylene Blue	Leuko-methylene blue	[122]
PAMAM G5	Au, Ag	Methylene blue	(a) Methylene blue oxide, (b) Phenol, benzenesulfonic acid	[121]
PAMAM G5	Au, Ag	Morin	(a) Substituted benzofuranone (morin oxide) (b) 2,4-dihydroxybenzoic acid, 2,4,6-trihydroxybenzoic acid	[115]
PAMAM G6	Pd/Au	Morin	(a) Substituted benzofuranone (morin oxide) (b) 2,4-dihydroxybenzoic acid, 2,4,6-trihydroxybenzoic acid	[118]
PAMAM-OH G6	Pd/Au	Methyl Orange	(a) Sulfanilic acid N,N-dimethyl, p-phenyldiamine (b) Benzene sulfonic acid, aniline	[114]
PAMAM G5 functionalized by maleic anhydride and cysteamine	Pd	Morin	(a) Substituted benzofuranone (morin oxide) (b) 2,4-dihydroxybenzoic acid, 2,4,6-trihydroxybenzoic acid	[120]
PAMAM G5 functionalized by maleic anhydride and cysteamine	Pd	Morin	(a) Substituted benzofuranone (morin oxide) (b) 2,4-dihydroxybenzoic acid, 2,4,6-trihydroxybenzoic acid	[119]

**Table 4 nanomaterials-12-00445-t004:** Dendritic Polymer-Metal NPs formulations for heterogeneous catalysis of nitroaromatics.

Dendritic Polymer	Metal NPs	Substrate-Formulation	Reference
PPI-G2	Au	Poly(4-vinyl pyridine) beads	[123]
PAMAM G4 Dendrons	Pt	SBA-15 SiO_2_	[128]
PAMAM G5 Dendrons	Ag	Polystyrene microsphere core SiO_2_ shells	[124]
PAMAM G7 Dendrons	Ag	Fe_3_O_4_ coated by polystyrene	[130]
PAMAM G3 Dendrons	Ag	Graphite	[135]
PEI	Au	Polyacrylonitrile fiber	[126]
Amphiphilic PPI G2, G3	Ag, Pd	MWCNT	[137]
PAMAM G1	Ag	Fe_3_O_4_	[131]
PAMAM G2	Au	Polyacrylic acid/polyvinyl alcohol nanofibers	[127]
PEI Core Polystyrene dodecyl shell	Au	Polymer open-cellular elastic monolith	
PEI Core Poly(styrene-co-2-ethyl hexyl acrylate shell.	Pt	Copolymer with 2-ethylhexyl acrylate-poly(ethylene glycol) dimethacrylate open-cellular elastic monolith	[142]
Amphiphilic PEI	Au	Chloroform, toluene, or petroleum ether	[141]
PEI Core	Au	Polystyrene dodecyl shell	[144]
PAMAM G6	Au	Cellulose nanocrystals	[125]
PAMAM-G4	Au/Ag	Glass microreactors	[129]
PPI G2	Ag	MWCNT	[139]
PEI Core poly(styrene-co-2-Ethylhexyl acrylate) Shell	Pt	Copolymerization with 2-ethylhexyl acrylate-poly(ethylene glycol) dimethacrylate	[143]
PAMAM G4	Au	Dialysis membrane	[140]
SiO2 Dendrons	Ag	Fe_3_O_4_@SiO_2_@dendritic-SiO_2_-NH_2_-Ag	[133]
PAMAM G2 Dendrons	Au	Fe_3_O_4_, KH-570 glycidyl methacrylate divinylbenzene copolymer,	[132]
PAMAM G3 dendrons	Pd	Graphene oxide, MnFe_2_O_4_ NPs	[136]

**Table 5 nanomaterials-12-00445-t005:** Dendritic Polymer-Metal NPs formulations for heterogeneous catalysis of pigments.

Dendritic Polymer	Metal NPs	Substrate-Formulation	Dyes	Reference
PAMAM Dendrons	Ag/Au	GO	Methyl orange, Congo red	[145]
PAMAM G4	Ag	Cellulose nanofibril films	Rhodamine B	[150]
PAMAM Dendrons G0, G1, G2	Au	Fe_3_O_4_ core 4-methyl styrene-divinylbenzene glycidyl methacrylate shell beads	Rhodamine B	[161]
Hyperbranched PG	Ag, Au	Poly(styrene)-co-poly(vinyl benzene chloride) beads	Congo Red	[155]
PAMAM G4	Ag, Au	SiO_2_ nanospheres	Methylene blue	[158]
PAMAM-OH G5	Pd/Au	CeO_2_, NiO, Fe_2_O_3_, MnO_2_, SiO_2_, Co_3_O_4_	Morin	[148]
PPI G2, G3	Au, Au/Pd	Poly(vinyl imidazole) microbeads	Malachite green	[157]
PPI(G2) and PPI(G3)	Ag, Au, Pd	4-Vinyl pyridine) beads	Trypan blue	[156]
PAMAM	Cu, Ag	Polyester nonwoven fabrics	4-Nitrophenol, methylene blue, malachite green, remazol red	[151]
PAMAM-OH G6	Pt	Mesoporous Co_3_O_4_	Methylene blue	[149]
PAMAM G1	Fe	Polyester fabrics	4-Nitrophenol, methylene blue	[154]
PEG	Fe	Fibrous polyester membrane	Malachite green	[152]
PAMAM	Fe	GO, polyester textile	Crystal violet	[153]
PAMAM, PEG dendrigraft	Au	Nanographene oxide	4-Nitrophenol, 4-nitroaniline Congo red	[146]
PAMAM G4	Au	SiO_2_ nanospheres	4-Nitrophenol, rhodamine B	[160]

**Table 6 nanomaterials-12-00445-t006:** Unconventional Dendritic Polymer formulations.

Dendritic Polymer	Active Ingredient	Substrate-Formulation	Pollutants	Reference
Hexa and nona-functionalized amido dendrons	Cu^2+^	Chitosan microbeads	Organic water and soil pollutants	[166]
PEI	Fe^3+^	Polyacrylonitrile fiber	Reactive red 195	[165]
Dendritic linear dendritic copolymer polybenzyl ether G2, G3, G4-PEG5000-G2	Laccase	-	Bisphenol A	[172]
PAMAM G3	Fe_3_O_4_	Montmorillonite or rice-straw-ash	Malachite green, xylenol orange	[167]
PEI	SiO_2_@TiO_2_ core-shell nanospheres	-	4-Nitrophenol	[168]
PEI	CeO_2_@SiO_2_ core-shell nanospheres	-	4-Nitrophenol	[169]
Hyperbranched polyamide	CuFe_2_O_4_	-	Reactive red 2, reactive yellow 3, basic red 46, basic yellow 24	[170]
Hyperbranched phenylene diamine methyl methacrylate	FeMnO_3_	-	Methylene blue, tetracycline, Rhodamine B	[171]
PAMAM G4	Hyaluronidase	-	Hyaluronic acid	[173]

**Table 7 nanomaterials-12-00445-t007:** Dendritic polymer photocatalyst formulations for aqueous pollutant disintegration.

Dendritic Polymer	Photocatalyst/Substrate	Pollutants	Reference
PAMAM-NH_2_ G4 PAMAM-OH G4 PAMAM-COONa G4	TiO_2_	2,4-Dichlorophenoxyacetic acid	[176]
Alkylated PAMAM G1, G2, G3 Dendrons	TiO_2_@SiO_2_ core-shell	2,4-Dichlorophenoxyacetic acid	[177]
Hyperbranched polyimide N-oxide (PINO)	-	Methyl orange	[197]
Aromatic Polyamide Dendrimer Functionalized with Spirolactam	TiO_2_	Phenol	[178]
Hyperbranched polyester HPES-OH	ΤιO_2_ nanowires	Wastewater	[179]
PAMAM dendrons	Fe_3_O_4_, TiO_2_,	Methyl orange	[180]
Poly (aryl benzyl ether) Zn phthalocyanine Dendrimer	Polycrystalline TiO_2_	Rhodamine B	[183]
PAMAM Dendron	CdS	Rhodamine B	[186]
PAMAM G4	ZnO	Naphthol blue black	[194]
PAMAM G4	TiO_2_/nanoclay polyester Fabric	Reactive red 4	[184]
PAMAM dendrons	Fe_3_O_4_, TiO_2_	Methyl orange	[181]
PAMAM G2.5	ZnO/CuS low-dimensional heterostructured composites	Rhodamine B, methylene blue	[192]
PAMAM G4	K4 [SiW12O40] (POM)	Methyl Red, methylene blue, alizarin yellow R, xylenol orange	[189]
PAMAM G0, G1, PEI	Pt TiO_2_	Brilliant black	[185]
PAMAM G0, G1 Dendrons	TiO_2_, CdS, Fe_3_O_4_	Methyl orange	[182]
Hyperbranched Polyester	ZnO	(N-(1-naphthyl)-ethylene diamine dichloride, catechol, sulfanilamide	[193]
Hyperbranched polyurethane	Reduced carbon dot-ZnO_2_	Dodecyl-benzenesulfonate, Commercial detergent	[195]
PAMAM G3	CeO_2_ pullulan/poly(vinyl alcohol)/poly(acrylic acid) nanofibers	Phenol, 4-hydroxy-1-naphthalene sulfonic acid sodium salt, and azorubine dye	[188]
PEI	BiVO4	Triclosan (5-chloro-2-(2,4-dichloro phenoxy)phenol)	[191]
Hyperbranched PAMAM	CuS reduced GO, dDialdehyde cellulose fiber	Rhodamine B	[187]
PAMAM	MoS_2_	Chlorpyrifos, glyphosate	[190]

## Data Availability

Not applicable.

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
