# Peer review of "Catalytic Neutralization of Water Pollutants Mediated by Dendritic Polymers"

_nanomaterials, 2022, doi:10.3390/nano12030445_

Round 1
Reviewer 1 Report
The manuscript "Catalytic Neutralization of Water Pollutants Mediated by Dendritic Polymers"summarized the simple and composite catalysts and photo-catalysts containing dendritic polymers and applied in water remediation, jointly with some unconventional solutions and future prospects. Overall, the article is well organized and its presentation is good. I strongly recommend that the paper be published in the nanomaterial.

Author Response
Thank you very much for your kind and encouraging comments.
Reviewer 2 Report
This work is applied in many environmentally benign implementations. One of the most important among them is water purification though pollutant decomposition. Simple and composite catalysts and photo-catalysts containing dendritic polymers and applied in water remediation will be discussed jointly with some unconventional solutions and future prospects. However, some critical issues remain to be solved and a thorough revision was needed:
- In section 2 for the reduction nitrophenols: Cite these refs.: J. Mol. Liq. 326 (2021) 115223; J. Mol. Liq. 330 (2021) 115522.
- In section 2.2. Treatment of pigments: Cite these refs.: J. Mol. Liq. 244 (2017) 226-232.
- In section 3.2. Heterogenous catalytic degradation of dyes: Cite these refs.: J. Mol. Struct. 1210 (2020) 128029; J. Mol. Struct. 1217 (2020) 128361; J. Mol. Liq. 344 (2021) 117082.
- In section 5.1. Photocatalysis on the Prominent TiO2 substrate: Cite these refs.: J. Mol. Struct. 1200 (2020) 127115.
- The manuscript needs thorough revision to improve the text quality and readability of the work.
Author Response
The relevant references were incorporated in the manuscript and thorough revision to improve the text quality and readability has been performed. We thank the reviewer for his helpful comments.
Reviewer 3 Report
The review article titled "Catalytic Neutralization of Water Pollutants Mediated by Dendritic Polymers" discusses the methods used for the synthesis of catalysts that are based on nanoparticles encapsulated in dendrimers, and compare their efficacy specifically in water decontamination processes. It is a nice review article and will attract broad audience. I would recommend to accept after a minor modification.
Some comments are below:
- May be some details for water pollutant degradation or related reactions will be helpful with suitable literature citations.
- The comparison table can be extended for other types of pollutants as well. It mostly says Morin, methylene blue and methylene orange dyes. these are not actual water pollutant, model dyes.
Author Response
We thank the reviewer for his constructive comments
- Figures 11, 18 and 25 contain degradation paths for Morin, Methyl Orange, Methylene Blue, Trypan Blue, Congo Red, chlorpyrifos and glyphosate.
- Tables 6 and 7 contain data for conventional pollutants: wastewater, herbicides (2,4-dichlorophenoxyacetic acid), pesticides (chlorpyrifos, glyphosate), antibiotics (Tetracycline), surfactans (commercial detergent, dodecyl-benzenesulfonate), biocides (Triclosan) e.t.c.
Reviewer 4 Report
This review entitled "catalytic neutralization of water pollutants mediated by dendritic polymers" is very interesting to read. It is really an important contribution to the field. I recommend publication of this review also for pedagogic reasons it wold be a perfect content for a graduate/PhD students working on chemistry and applications of nanomaterials.
But as I feel I am not a total expert on the dendritic polymers domain, I recommend this submitted paper should be given to an expert in those materials for an exhaustic reviewing.
Author Response
We are grateful to the reviewer for his polite and very encouraging comments